# Research

Subject Areas:
energy/engineering geology/environmental engineering

Keywords:
bedding, principal strain difference, volumetric strain, permeability, lateral expansion coefficient

Author for correspondence:
Chao Liu
e-mail: cquliuchao@cqu.edu.cn

# Influence of principal stress effect on deformation and permeability of coal containing beddings under true triaxial stress conditions

Jiahui Dai[1,2,3], Chao Liu[1,2], Minghui Li[1,2,4] and Zhenlong Song[1,2]

[1]State Key Laboratory of Coal Mine Disaster Dynamics and Control, and [2]College of Resources and Environmental Science, Chongqing University, Chongqing 400030, People's Republic of China
[3]China Coal Technology Engineering Group Chongqing Research Institute, Chongqing 400037, People's Republic of China
[4]State Key Laboratory for GeoMechanics and Deep Underground Engineering, China University of Mining and Technology, Xuzhou 221116, People's Republic of China

CL, 0000-0002-4213-7000

*In situ* stress is generally an anisotropic/true triaxial stress ($\sigma_1 > \sigma_2 > \sigma_3$). Bedding weakens the continuity and integrity of coal. It is critical to understand the mechanical behaviour and gas migration of coal under true triaxial stress conditions. We performed experiments of cubic coal samples to investigate the permeability evolution and mechanical behaviour of coal under true triaxial stress conditions by using newly developed true triaxial geophysical apparatus. We analysed the effect of principal stresses on deformation and permeability characteristics of coal containing bedding planes. The results show that volumetric strain, stress states and bedding directions determine the permeability comprehensively. The variable quantity of strain was the largest in the direction normal to the bedding plane. The expansion or compression degree was characterized by the difference between the major and minor principal strain ($\varepsilon_1 - \varepsilon_3$). Essentially, this represents the difficulty degree with regard to coal being compressed at the initial stress state and the deformation degree in $\varepsilon_1$ and $\varepsilon_3$ direction. The variation of ($\varepsilon_1 - \varepsilon_3$) was consistent with that of permeability. Under an identical true triaxial stress condition, permeability was smaller when larger stress was applied in the direction normal to the bedding plane. Additionally, stress level in the direction parallel to the bedding planes and the directions between stresses in the direction parallel to the bedding

planes and the flow direction also affect the permeability and strain. By solving lateral expansion coefficient, coal also exhibited anisotropic properties.

# 1. Introduction

In the process of underground mining and tunnelling, coals are actually under true triaxial stress conditions ($\sigma_1 > \sigma_2 > \sigma_3$) due to the combined effects of tectonic stress, mine-induced stress and roadway arrangement forms [1]. It is well known that coals are sedimentary rocks with bedding planes. Bedding and their orientation have significant influences on reservoir permeability and induce significant anisotropy in mechanical behaviour and failure strength [2,3]. In addition, the relative positions among the directions of beddings and each principal stress are uncertain. Coal seams contain abundant pores, cracks and two sets of cleats (perpendicular to the bedding) [4], which jointly determine the obvious directivity of displacement and permeability of coal. Under the disturbance of mine-induced stress, local blasting and adjacent coal pillars, the gas in the coal seams can easily lead to energy instability, and cause coal and gas outbursts. Therefore, it was of significant importance to study the deformation characteristics of raw coal containing beddings, and the gas migration properties at the action of the external force under true triaxial stress states.

Koenig & Stubbs conducted permeability experiments on coals and the permeability ratio of different bedding directions reached a maximum of $17 : 1$ [5]. Pan & Connell took coal with different bedding directions as the research object and studied the law of the permeability evolution during the loading process. When the gas drainage borehole was arranged along the coal seam, the effect was better [6]. Szwilski concluded that the elastic modulus is greater in the direction parallel to the bedding planes than that normal to the bedding planes [7]. Day *et al.* concluded that swelling strains in the two directions parallel to the bedding planes were almost identical, while the swelling in the plane perpendicular to the bedding plane was higher than that parallel to the bedding [8]. Liang *et al.* conducted uniaxial compression tests on salt rock and coal specimens to obtain relevant basic mechanical parameters and strength characteristics of salt rock and coal in different bedding directions [9]. Huang & Liu analysed the effect of bedding plane properties and stress state on fracture propagation under true triaxial stress, and established three-dimensional propagation models [10]. Alexeev *et al.* conducted a true triaxial compression experiment on a rock specimen, and concluded that it was fractured at the maximum dilatation of the rock [11]. Pan *et al.* studied the influence of intermediate principal stress ($\sigma_2$) on rock failure by the EPCA3D system. It was concluded that when $\sigma_2$ was moderated, it could hinder the local rock damage while increasing rock strength. Additionally, when $\sigma_2$ reached a certain value, it formed local damage [12]. Cai discussed the influence of $\sigma_2$ on rock splitting and rock strength near the excavation boundary by using the FEM/DEM numerical analysis tool, and concluded that higher $\sigma_2$ restricted the occurrence of micro-cracks and fractures parallel to the $\sigma_1$ and $\sigma_2$ directions [13]. Li *et al.* studied the mechanical responses of rock by unloading $\sigma_3$ and loading $\sigma_1$ under different $\sigma_2$, and concluded that the rock failure mode may change from shear to slabbing when $\sigma_2$ increases to a critical value for strong and hard rocks [14].

In combination with an actual engineering background and by designing relevant stress paths, many scholars have studied the permeability evolution law. Based on the practical engineering background of protective layer mining, Yin *et al.* investigated the mechanical and permeability characteristics of raw coal by increasing axial compression while decreasing confining pressure, simultaneously. They found that the peaks of deviatoric stress and corresponding strain were smaller than those obtained in conventional triaxial compression experiments [15]. In combination with CT scanning, Chen *et al.* studied the damage and permeability of briquette from effective confining pressure during the unloading process. In comparison with the experiment of unloading confining pressure under fixing deviatoric stress, where the effective confining pressure was unloaded from 8 MPa to 6 MPa, the damage variables increased by 0.084, and permeability increased by 16.7% [16]. Gentzis *et al.* found that permeability measured perpendicular to the bedding planes was three to four times lower than that measured parallel to the bedding planes [17].

In summary, there have been relatively few studies on the deformation and permeability of raw coal containing beddings under true triaxial stress conditions. Based on this, by considering the relative positions of principal stress and bedding direction, the experiment of loading and unloading the intermediate principal stress while fixing the major and minor principal stresses was carried out. The differences among the permeability and deformation evolutions in all directions were discussed.

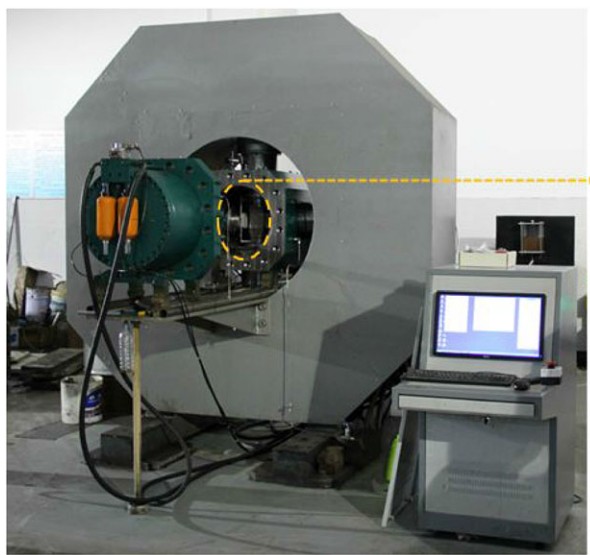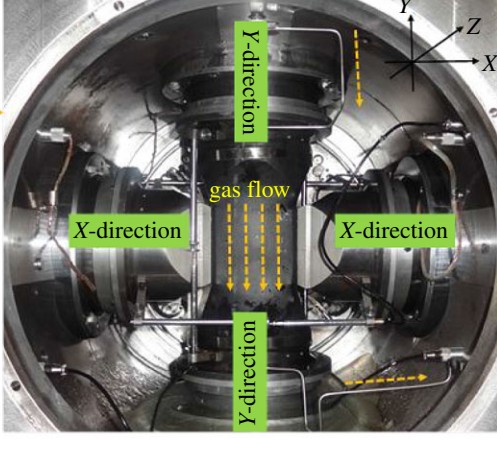

**Figure 1.** Multi-functional true triaxial geophysical apparatus.

The purpose of the research is to investigate the deformation mechanism of the coal and gas migration in various directions under the effect of bedding.

## 2. Experimental apparatus and scheme

### 2.1. Experimental apparatus

The experiments were conducted with the newly developed true triaxial geophysical (TTG) apparatus [1,18,19]. The system consisted of a frame-type stand, true triaxial pressure cell, loading system, internally sealed fluid flow system, measurement system, control and data acquisition system, and acoustic emission system. The apparatus could realize the experimental investigation of mechanical properties and seepage characteristics under true triaxial stress conditions. The experimental device and loading chamber are shown in figure 1.

### 2.2. Experimental specimens

The coals were obtained from the outburst coal seam C1 of the Baijiao coal mine in the Sichuan Coal Industry Group. They were drilled, cut and polished into $100 \times 100 \times 100$ mm cube specimens. The two ends of the cube specimen were finally ground to meet the requirements of ISRM-suggested methods for the parallelism of the end faces [20]. The prepared specimens can be seen in figure 2.

### 2.3. Experimental scheme

As shown in figure 2, in accordance with the bedding parallel to the $Y$-direction and $Z$-direction, and perpendicular to the $X$-direction, the specimen was placed into the TTG loading chamber. The four experimental conditions are described below.

Condition 1: $\sigma_X$, $\sigma_Y$ and $\sigma_Z$, were simultaneously increased to 50 MPa of hydrostatic pressure at the rate of 0.05 MPa s$^{-1}$ and maintained at this pressure for 10 min. Then, $\sigma_X$ and $\sigma_Z$ were reduced to 15 MPa at an identical rate. The discharge flow rate at the outlet was recorded at each step and was used to calculate permeability. For the sake of safety, the gas environment was $CO_2$ and the gas pressure was 3 MPa.

Condition 2: By keeping $\sigma_Y$ and $\sigma_Z$ unchanged, $\sigma_X$ was raised to 20, 25, 30, 35, 40, 45 and 50 MPa.

Condition 3: By keeping $\sigma_X$ and $\sigma_Z$ unchanged, $\sigma_Y$ was reduced to 45, 40, 35, 30, 25, 20 and 15 MPa.

Condition 4: $Z$-direction loading, $X$-direction unloading, $Y$-direction loading and $Z$-direction unloading, were continued as described in Conditions 2 and 3. When loaded or unloaded in one direction, the force was kept constant in the other two directions.

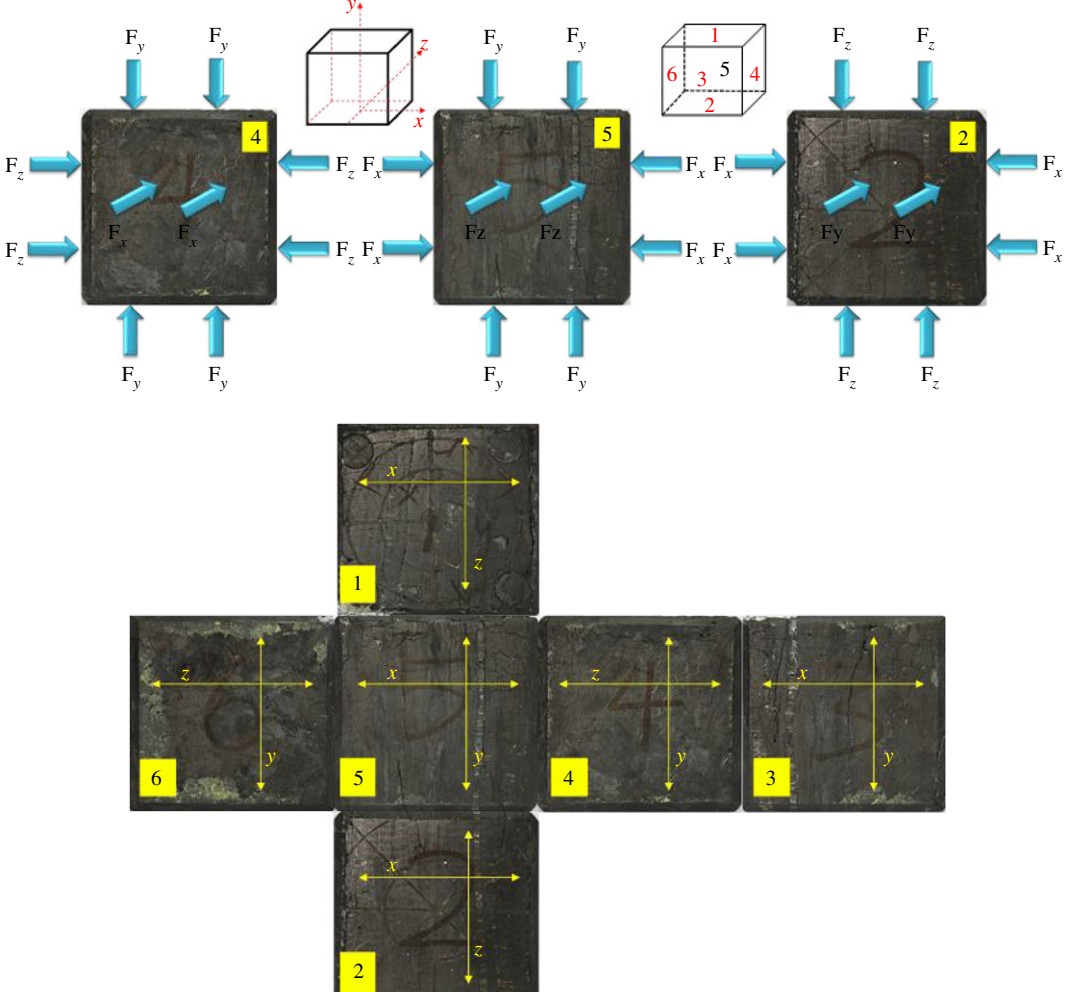

**Figure 2.** Coal specimens.

It is true that $CO_2$ adsorption will cause the coal matrix swelling [21,22]. Hence, the total strain and permeability of the coal are affected by $CO_2$ adsorption. However, it is very difficult to figure out the actual effects of $CO_2$ adsorption on the deformation and permeability quantitatively. Besides, in our experiments, the gas pressure was constant (3 MPa). During the whole permeability measurements, the inlet pressure was kept at 3 MPa and the outlet pressure kept at 0.1 MPa. As the gas pressure kept constant, the effects of $CO_2$ adsorption on the deformation and permeability kept constant as well. Therefore, the measured strain and permeability were under comprehensive conditions, which can reflect the real field situations.

Similarly, Darcy fluid flow occurs in the natural fracture system for coals [23], which is a result of flow in the cleat system, and its permeability was calculated as [1,18,24]

$$k = \frac{2q\mu L P_2}{[A(P_1 + P_2)(P_1 - P_2)]},$$ (2.1)

where $k$ is the permeability in $m^2$; $q$ is the exit flow rate of $CO_2$ in $m^3\,s^{-1}$; $P_2$ is one standard atmospheric pressure in MPa; $\mu$ is the $CO_2$ kinematic viscosity of $CO_2$ in MPa s, at the temperature of the test according to Sutherland's formula; $L$ is the specimen length in m; $P_1$ is the entrance pressure in MPa, of the test's $CO_2$ at test temperature; and $A$ is the cross-sectional area of the coal specimens in $m^2$.

The relationship between principal stress in each direction and the major (or minor and intermediate) principal stress is shown in table 1. The stress path is shown in figure 3. The relationship between stress and bedding is shown in figure 4.

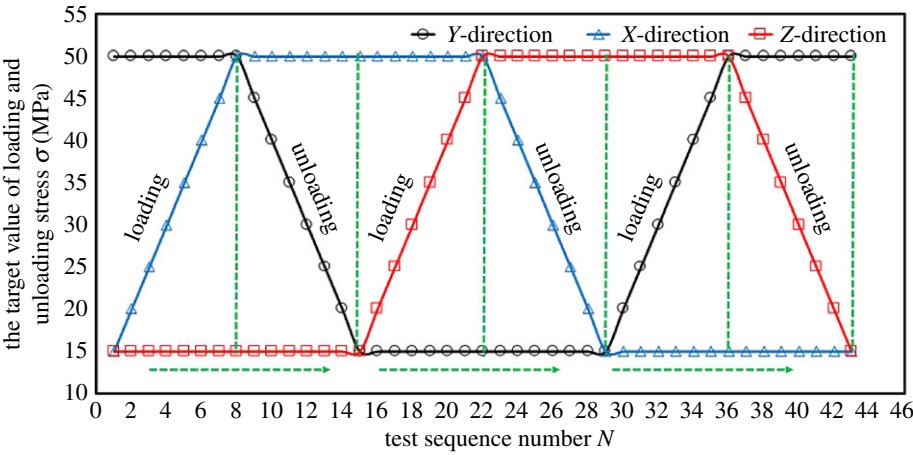

**Figure 3.** Experimental stress path.

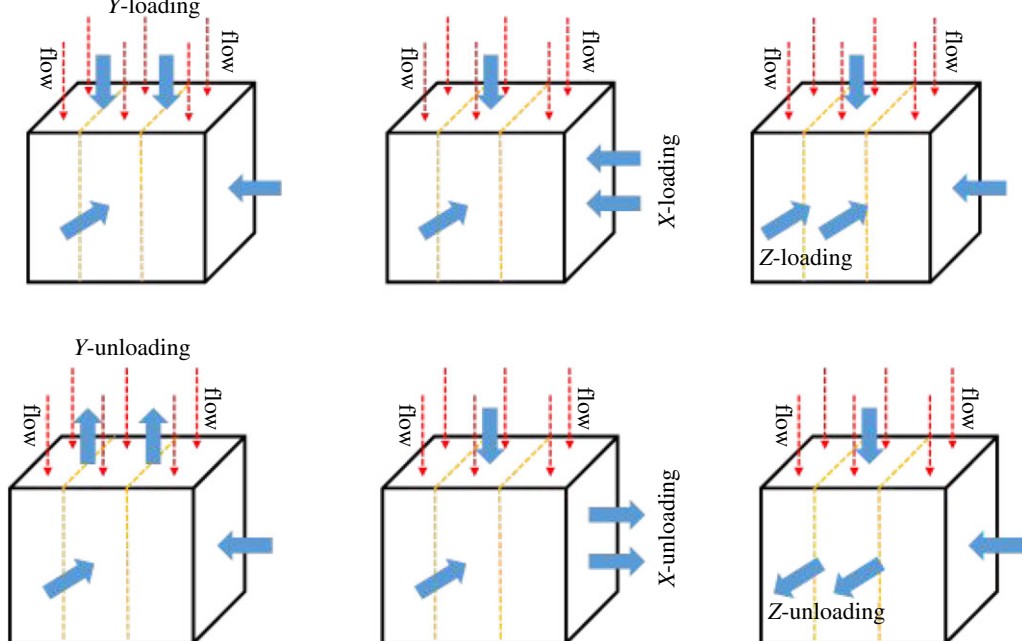

**Figure 4.** The stress state and schematic diagram of the relationship between stress and bedding.

**Table 1.** The relationship between the direction of X, Y, Z, and $\sigma_1$, $\sigma_2$, $\sigma_3$.

| experimental sequence | loading mode | $\sigma_1$ | $\sigma_2$ | $\sigma_3$ |
|---|---|---|---|---|
| 1 | X-direction loading | Y | X | Z |
| 2 | Y-direction unloading | X | Y | Z |
| 3 | Z-direction loading | X | Z | Y |
| 4 | X-direction unloading | Z | X | Y |
| 5 | Y-direction loading | Z | Y | X |
| 6 | Z-direction unloading | Y | Z | X |

## 3. Results and analysis

When coals were compressed in one direction, the strain value showed an increasing tendency. On the contrary, it decreased during the expansion process. Figure 5 shows the variation of $\varepsilon_V$ and $k$ during the

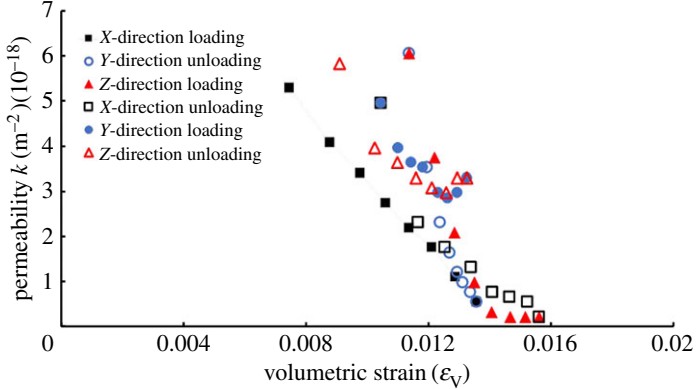

**Figure 5.** Variation of volumetric strain $\varepsilon_V$ and permeability $k$ during the entire process.

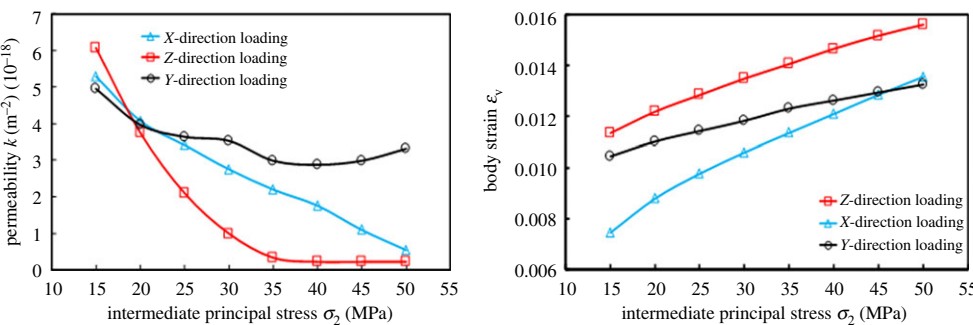

**Figure 6.** Variation of permeability $k$ and strain $\varepsilon_V$ with intermediate principal stress $\sigma_2$ loading.

entire experimental process. $\varepsilon_V$ and $k$ basically comply with the following rules: with the increase of $\varepsilon_V$, a decrease in $k$ induced by the compaction and closure of micro-porosities and microcracks occurred, and vice versa. When $\sigma_2$ was in relatively high-stress levels, $k$ increased as $\sigma_2$ increased. This was inconsistent with the above phenomena, and this part will be analysed in detail.

## 3.1. Analysis of permeability

According to the stress-designed path, the stress starting points were $\sigma_1 = 50$ MPa, $\sigma_2 = \sigma_3 = 15$ MPa, $\sigma_1 = \sigma_2 = 50$ MPa and $\sigma_3 = 15$ MPa, respectively. Regarding the stress loading and unloading starting points as the base points, the variation of strain ($\Delta\varepsilon$) can be defined as

$$\Delta\varepsilon_1 = \varepsilon_{1-i} - \varepsilon_{1-15}, \tag{3.1}$$

$$\Delta\varepsilon_2 = \varepsilon_{2-i} - \varepsilon_{2-15} \tag{3.2}$$

and
$$\Delta\varepsilon_3 = \varepsilon_{3-i} - \varepsilon_{3-15}, \tag{3.3}$$

where $\varepsilon_{1-15}$, $\varepsilon_{2-15}$ and $\varepsilon_{3-15}$ are the major, intermediate and minor principal strain under the condition of $\sigma_2 = 15$ MPa, respectively. $\varepsilon_{1-i}$, $\varepsilon_{2-i}$ and $\varepsilon_{3-i}$ are the major, intermediate and minor principal strain under the condition of $\sigma_2 = i$ MPa ($i = 15, 20, 25, 30, 35, 40, 45$ and $50$), respectively. $\Delta\varepsilon_1$, $\Delta\varepsilon_2$ and $\Delta\varepsilon_3$ are the variation of major, intermediate and minor principal strain.

Figure 6 shows the variation of $k$ and $\varepsilon_V$ with $\sigma_2$ loading. $\varepsilon_V$ was $\varepsilon_{VZL} > \varepsilon_{VYL} > \varepsilon_{VXL}$ (ZL, YL and XL represent the loading of Z-direction, Y-direction and X-direction, respectively.). Therefore, the compression degree of the specimen was the largest in ZL, while the YL was the second largest, and the XL was the smallest.

Since the absolute value $|\Delta\varepsilon|$ meets $|\Delta\varepsilon_{1ZL}| < |\Delta\varepsilon_{1YL}|$, $|\Delta\varepsilon_{2ZL}| > |\Delta\varepsilon_{2YL}|$ and $|\Delta\varepsilon_{3ZL}| < |\Delta\varepsilon_{3YL}|$, the volume of the coal specimen was smaller during the Z-direction loading process. Since $\sigma_1$ was perpendicular to bedding, the available compression space in this stress arrangement was relatively reduced during the ZL process. As the Z-direction continued to be loaded, the coal became more compressed, in which case $\varepsilon_V$ was determined to be the largest. In other words, its volume was minimal.

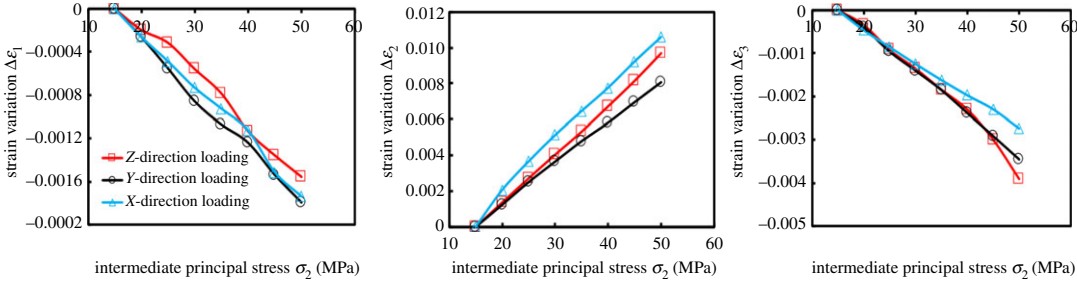

**Figure 7.** Strain variation $\Delta\varepsilon$ with intermediate principal stress $\sigma_2$ loading.

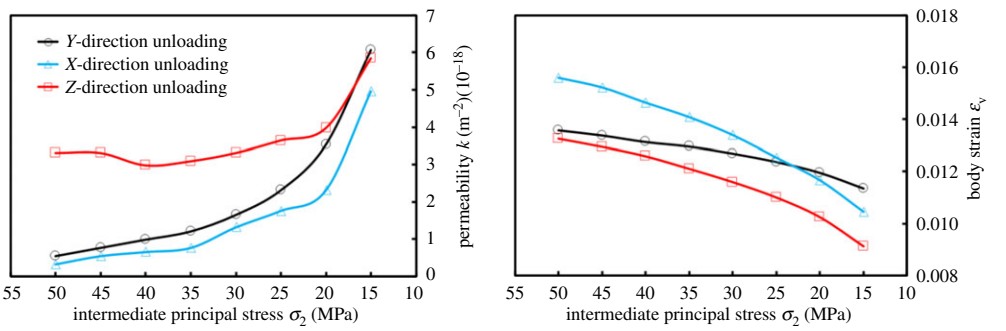

**Figure 8.** The variations of permeability $k$ and strain $\varepsilon_V$ with intermediate principal stress $\sigma_2$ unloading.

Although $|\Delta\varepsilon|$ meets $|\Delta\varepsilon_{1YL}| > |\Delta\varepsilon_{1XL}|$, $|\Delta\varepsilon_{2YL}| < |\Delta\varepsilon_{2XL}|$ and $|\Delta\varepsilon_{3YL}| > |\Delta\varepsilon_{3XL}|$, $|\Delta\varepsilon_{1YL}| + |\Delta\varepsilon_{3YL}| - |\Delta\varepsilon_{2YL}| < |\Delta\varepsilon_{1XL}| + |\Delta\varepsilon_{3XL}| - |\Delta\varepsilon_{2XL}|$. Based on this, the compression degree was superior in YL, in comparison to loading in the X-direction. Since the X-direction was perpendicular to bedding, the bedding bearing capacity was weaker, and the deformation response was obvious under the influence of the external force, and the pores and cracks were mainly compacted, which led to denser coal. Therefore, the result showed that $|\Delta\varepsilon_{2YL}| < |\Delta\varepsilon_{2XL}|$. Similarly, since $\sigma_3$ was perpendicular to bedding when loaded in the Y-direction, a larger amount of expansion could be produced as shown in $|\Delta\varepsilon_{3YL}| > |\Delta\varepsilon_{3XL}|$. In the $\sigma_1$-direction, the swelling degree between the two was very small.

During the loading process, $k_{ZL}$ was always minimum, and the relationship between $k_{XL}$ and $k_{YL}$ was more complex. When loaded in the Z-direction, both $\varepsilon_V$ and the amount of compressed space were largest, and exhibited minimum $k_{ZL}$. Rock strength firstly increases and subsequently decreases with the increase of intermediate principal stress [12]. When the intermediate principal stress increases to the critical value, the resistance to deformation of coals is weakened, which makes the pores and cracks in coal body tend to open from the original closed state, and finally show an increase in permeability. Therefore, when loaded from 45 to 50 MPa, permeability increased. $\sigma_1$ and $\sigma_2$ were applied to bedding in parallel, where the coal was most likely to fail. When $\sigma_2$ was at a lower stress level, $k$ decreased with increasing $\sigma_2$. With a constant loading of $\sigma_2$, the coal entered a yield state that more micro-cracks initiate and coalesce [25], and correspondingly $k$ began to increase, which does not contradict the coals being in a compressed state.

When loaded in the X-direction, the coal did not tend to yield or fail, therefore, $k$ always decreased in this situation (figure 7).

Figure 8 shows the variation of $k$ and $\varepsilon_V$ with $\sigma_2$ unloading. $\varepsilon_V$ was generally $\varepsilon_{VXUL} > \varepsilon_{VYUL} > \varepsilon_{VZUL}$ (ZUL, YUL and XUL represent the unloading of Z-direction, Y-direction and X-direction, respectively); that is, the swelling degree was largest when unloaded in the Z-direction, while being the second largest in the Y-direction and smallest in the X-direction. $k$ meets $k_{ZUL} > k_{YUL} > k_{XUL}$.

When unloaded in the Y-direction and Z-direction, respectively, $\sigma_1$ and $\sigma_3$ are perpendicular to bedding. Therefore, the absolute value of $\Delta\varepsilon$ was $|\Delta\varepsilon_{1YUL}| > |\Delta\varepsilon_{1ZUL}|$ and $|\Delta\varepsilon_{3ZUL}| > |\Delta\varepsilon_{3YUL}|$. The dilatancy probability was larger under the ZUL, such that $|\Delta\varepsilon_{2ZUL}| > |\Delta\varepsilon_{2YUL}|$.

For YUL and XUL, it was shown that $|\Delta\varepsilon_{2XUL}| > |\Delta\varepsilon_{2YUL}|$, since $\sigma_2$ was perpendicular to bedding. During the YUL process, since $\sigma_1$ was perpendicular to bedding, this caused the coal to produce a superior amount of compression in the X-direction. However, when it was unloaded in the X-direction, the bearing capacity was weaker and a greater amount of compression was also generated in the

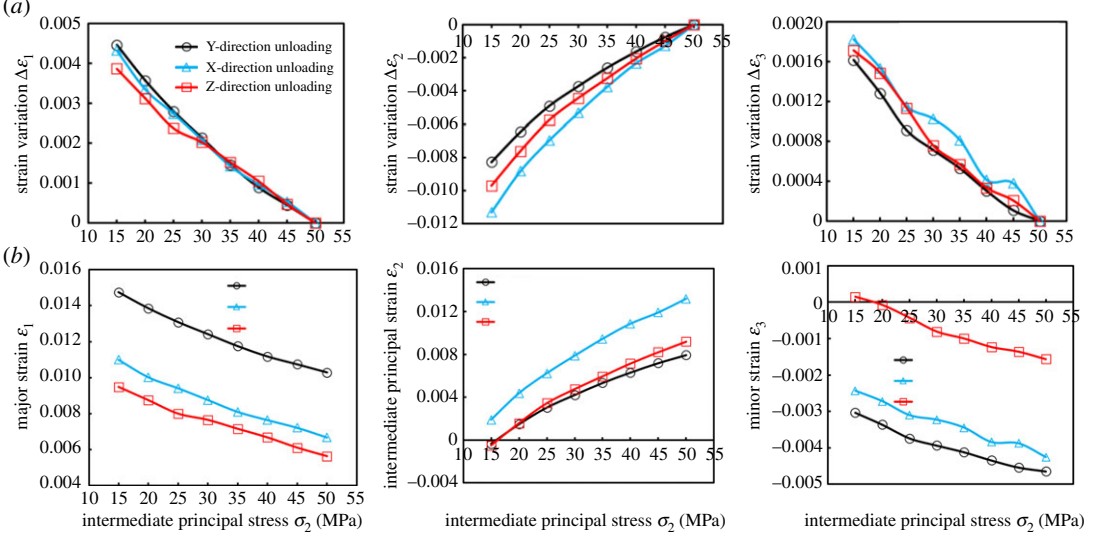

**Figure 9.** Strain $\varepsilon$ and strain variation $\Delta\varepsilon$ versus intermediate principal stress $\sigma_2$.

Z-direction due to $\sigma_1$ being parallel to bedding, in comparison to the stress arrangement in which $\sigma_1$ was perpendicular to bedding, a destructive tendency could easily be developed. This paper argues that if the stress level of $\sigma_1$ was increased further, or if $\sigma_3$ was reduced further, the X-directional compression degree in YUL may be less than in the Z-directional compression degree in XUL. Similarly, when unloaded in the X-direction, the coal produces more pores and cracks under the action of $\sigma_1$, which makes it more 'fragmentized'. In other words, its integrity was weakened. In order to maintain $\sigma_3$ unchanged and coals stabilized, it was necessary to supplement a larger amount of compression. Therefore, the ultimate manifestation was $|\Delta\varepsilon_{3XUL}| > |\Delta\varepsilon_{3YUL}|$.

The variation of $k$ basically accords with common sense, by which, the bigger the compression degree was, the smaller was the permeability during the unloading process. When $\sigma_2$ was unloaded to 20 and 15 MPa, $\varepsilon_V$ was $\varepsilon_{VYUL} > \varepsilon_{VXUL}$, and $k_{XUL} > k_{YUL}$ did not exist in figure 8. However, when unloaded from 20 to 15 MPa, the growth rate of $k$ was 71.88%; however, it was 114.29% in XUL. In this study, the cause of $k_{XUL} < k_{YUL}$ was considered as follows: (1) with consideration of the flow detection period, $k$ had a certain time difference with respect to deformation; that is, the flow variation exhibited hysteresis; (2) the target value of stress was limited. When $\sigma_2$ was unloaded to a value that was less than 15 MPa, then, $k_{XUL} > k_{YUL}$. In comparison with the growth rate of $k$, the probability of this phenomenon was larger. In addition, this also demonstrated that $\varepsilon_V$ and bedding simultaneously affected the variation of permeability and its trend.

It is worth noting that the expression of $\varepsilon_V$ was $\varepsilon_V = \varepsilon_1 + \varepsilon_2 + \varepsilon_3$ ($|\varepsilon_1| > |\varepsilon_2| > |\varepsilon_3|$). The variation of $\varepsilon_V$ with $\sigma_2$ may be dominated by the variation of $\varepsilon_1$. When $|\varepsilon_1|$ had a larger value at the initial stress point $(\sigma_1, \sigma_2, \sigma_3) = (50, 15, 15 \text{ MPa})$ or $(\sigma_1, \sigma_2, \sigma_3) = (50, 50, 15 \text{ MPa})$, the strain variation in the $\varepsilon_1$-direction obscured the variation of $\varepsilon_2$ and $\varepsilon_3$. Therefore, the variation of $\varepsilon_V$ was controlled by $\varepsilon_1$; that is, it was administrated by a larger value. When the strain was larger at the initial stress point, this means that the coals had been compressed to a certain extent, and that $k$ and its variation degree may have been relatively smaller under the same stress path. From figure 9b, $\varepsilon_1$ was $|\varepsilon_{1YUL}| > |\varepsilon_{1XUL}| > |\varepsilon_{1ZUL}|$. In the case of unloading, $\varepsilon_V$ was always $\varepsilon_{VXUL} > \varepsilon_{VYUL} > \varepsilon_{VZUL}$, such that the variation of $\varepsilon_V$ was determined by the principal strain in all directions, which belongs to process variation. Owing to the different sensitivity of $k$ to the deformation in each direction, the controlling effect of $k$ with a smaller deformation in one direction was larger than that in the other directions, and exhibited different $k$ at the macro level. Therefore, $\varepsilon_V$ was an important factor affecting $k$ rather than the determinant, as shown in figure 6.

## 3.2. Analysis of permeability variation amount $\Delta k$

Similarly, the permeability variation amount ($\Delta k$) was defined as

$$\Delta k = k_i - k_{15},\tag{3.4}$$

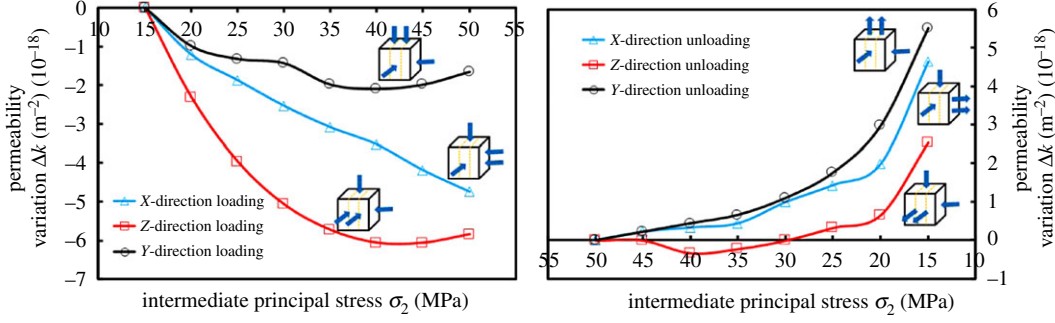

**Figure 10.** Variations of permeability variation $\Delta k$ with intermediate principal stress $\sigma_2$.

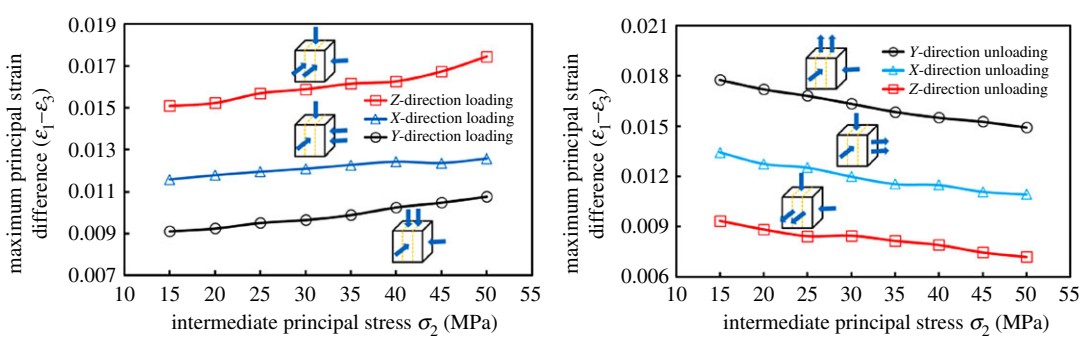

**Figure 11.** The variation of maximum principal strain difference $(\varepsilon_1 - \varepsilon_3)$ with $\sigma_2$ loading – unloading.

where $k_{15}$ is the permeability under the condition of $\sigma_2 = 15$ MPa, m². $k_i$ is the permeability under the condition of $\sigma_2 = i$ MPa, m².

As shown in figure 10, the variations of $\Delta k$ and $k$ were different with respect to loading and loading direction. $\Delta k$ meets $|\Delta k_{ZL}| > |\Delta k_{XL}| > |\Delta k_{YL}|$ in the loading process and when unloaded, and $\Delta k$ was $|\Delta k_{YUL}| > |\Delta k_{XUL}| > |\Delta k_{ZUL}|$. The end result of permeability variation was caused by permeability reduction that was incurred by compression in one direction, and that the permeability increase that was incurred by expansion in one direction was uncertain. In addition, the contribution degree of the permeability increase was also uncertain if the coal was in an expansion state in two directions. These conditions make the variations of $\Delta k$ and $k$ inconsistent. The author considers that $\sigma_2$-loading (or unloading) was an important determinant in the variation of $k$, and that the expansion or compression in the $\sigma_1$ and $\sigma_3$ directions determines the variation of $\Delta k$.

The bearing capacity was weaker during the loading process when $\sigma_2$ was parallel to the bedding plane, and coal was more likely to enter the yield state or cause vertical bedding separation [26], in comparison to XL in terms of the absolute value $|\Delta k|$.

Figure 11 shows the variation of the difference between the major principal strain and the minor principal strain $(\varepsilon_1 - \varepsilon_3)$ with $\sigma_2$ loading (or unloading). The variation of $(\varepsilon_1 - \varepsilon_3)$ was consistent with the variation of $\Delta k$ due to different loading and unloading directions. During the loading process, the coal in the $\varepsilon_1$ and $\varepsilon_3$ direction expanded, and $\varepsilon_1$ and $\varepsilon_3$ showed a decreasing trend in the numerical variations. The reasons for larger $(\varepsilon_1 - \varepsilon_3)$ were as follows: (1) the value of $\varepsilon_1$ did not have a significant difference throughout the loading process. However, the decreasing degree of $\varepsilon_3$ was relatively larger. This degree could be expressed by fitting the slope (m); (2) the strain value of the initial stress point (50, 15 and 15 MPa) was relatively larger, which indicates that the compression degree had been larger at the beginning of loading. With $\sigma_2$ loading, the amount of compression space could be decreased correspondingly.

The absolute slope value $(|m_{3-ZL}|)$ was maximum in the ZL process as shown in figure 12a. The results show that $|\Delta \varepsilon_3|$ was the largest; that is, the expansion amount in the Y-direction was the largest. In comparison to the XL and YL, the seepage path was longer and promoted the decrease of permeability in the ZL. When loaded in three directions, the slopes were substantially equal between $\varepsilon_1$ and $\sigma_2$, even if the initial value of $\varepsilon_1$ (the value of $\varepsilon_1$ at $\sigma_1 = 50$ MPa, $\sigma_2 = \sigma_3 = 15$ MPa) was larger when loaded in the Z-direction. The reasons were as follows: although the coal was in a relatively

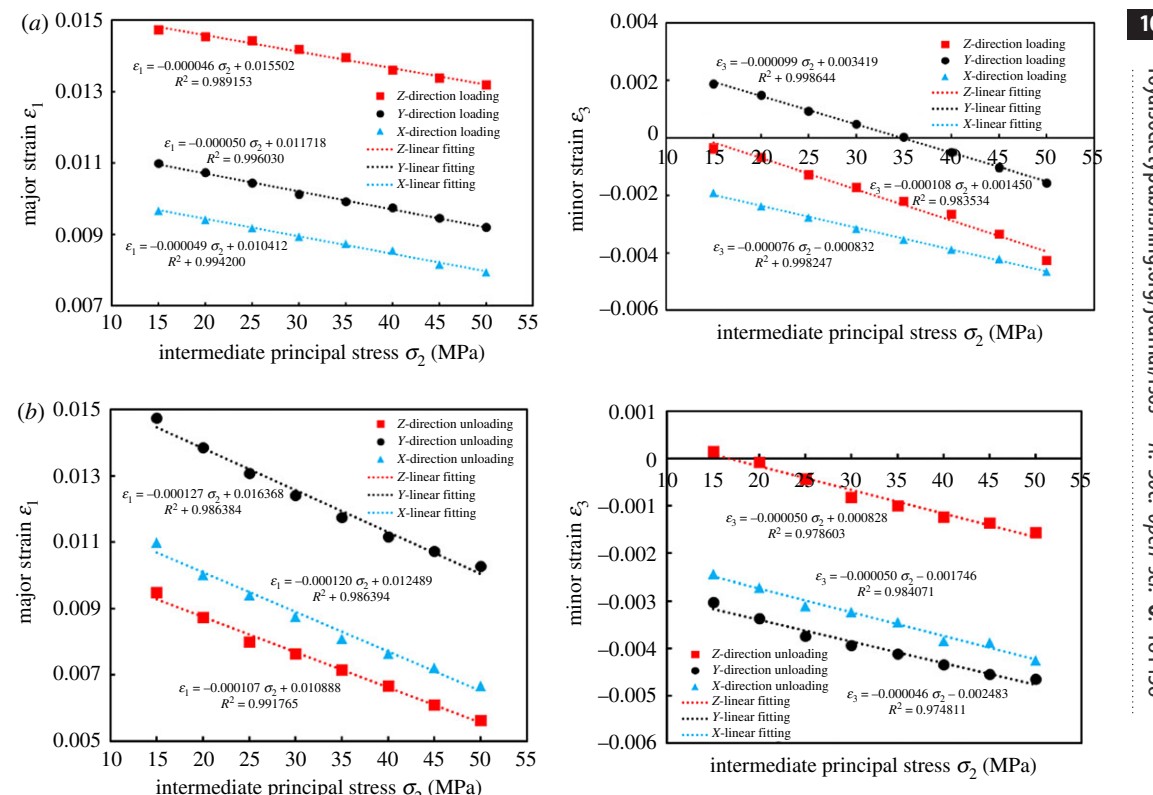

**Figure 12.** Variation of major strain $\varepsilon_1$ and minor strain $\varepsilon_3$ with intermediate principal stress $\sigma_2$ and its linear fitting results during loading and unloading process.

larger compression degree in the $X$-direction ($\parallel \varepsilon_1$), the expansion in this direction was substantially equal to that of the XL and YL, due to the fact that the $X$-direction was perpendicular to bedding.

For the analysis of $\Delta k$ incurred by the XL and YL, both slopes were almost equal, which indicated that the deformation amounts in the $\varepsilon_1$-direction were almost equal. However, the initial value of $\varepsilon_1$ was $\varepsilon_{1YL} > \varepsilon_{1XL}$, which indicated that the coals were compressed to a certain extent in the $\varepsilon_1$-direction when initially loaded in the $Y$-direction. Similarly, the initial value of $\varepsilon_{3YL} > \varepsilon_{3XL}$, and was also compressed up to a certain extent in the $\varepsilon_3$-direction. When loaded in the $Y$-direction, the space of pores and fractures that could be compressed was relatively smaller with $\sigma_2$ loading, and the macroscopic performance was smaller in $|\Delta k_{YL}|$. In conclusion, $\Delta k$ was $|\Delta k_{ZL}| > |\Delta k_{XL}| > |\Delta k_{YL}|$. The relevant variations were no longer analysed in the unloading process.

## 3.3. Analysis of permeability in different true triaxial stress states

As shown in figure 13, in order to further explore the properties of bedding, the permeability and deformation of six different combinations under the true triaxial stress state ($\sigma_1 = c$ MPa, $\sigma_2 = b$ MPa, $\sigma_3 = a$ MPa, $c > b > a$) were investigated.

Owing to typical non-isotropic properties, $k$ and $\varepsilon$ were significantly different under the six different stress states shown in figures 14 and 16. Equation (3.5) was the expression of the intermediate principal stress coefficient $b$, as follows:

$$b = \frac{\sigma_2 - \sigma_3}{\sigma_1 - \sigma_3}. \tag{3.5}$$

Different permeability relationships are shown in table 2 under six kinds of stress states.

When stress patterns were arranged in the form of, $1^{\#}$ and $2^{\#}$, $1^{\#}$ and $6^{\#}$, $2^{\#}$ and $5^{\#}$, $3^{\#}$ and $4^{\#}$, $3^{\#}$ and $6^{\#}$, and $4^{\#}$ and $5^{\#}$, from table 2 and figure 14, $k$ showed an identical change law. In other words, in the same true triaxial stress state, $k$ was smaller when a larger force was applied perpendicularly to the bedding plane. When the stress distributions were $1^{\#}$ and $4^{\#}$, $2^{\#}$ and $3^{\#}$, and $5^{\#}$ and $6^{\#}$, $k$ exhibited obvious anisotropy. When maintaining $\sigma_X = \sigma_\perp$ unchanged and $\sigma_{\parallel 1} > \sigma_{\parallel 2}$, the effect of $(\sigma_Y, \sigma_Z) = (\sigma_{\parallel 1}, \sigma_{\parallel 2})$ and

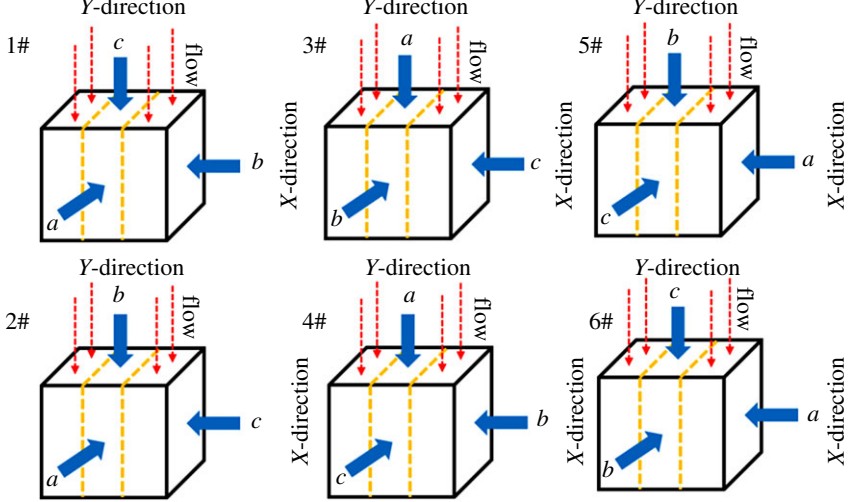

**Figure 13.** Six different stress arrangement combinations under true triaxial condition.

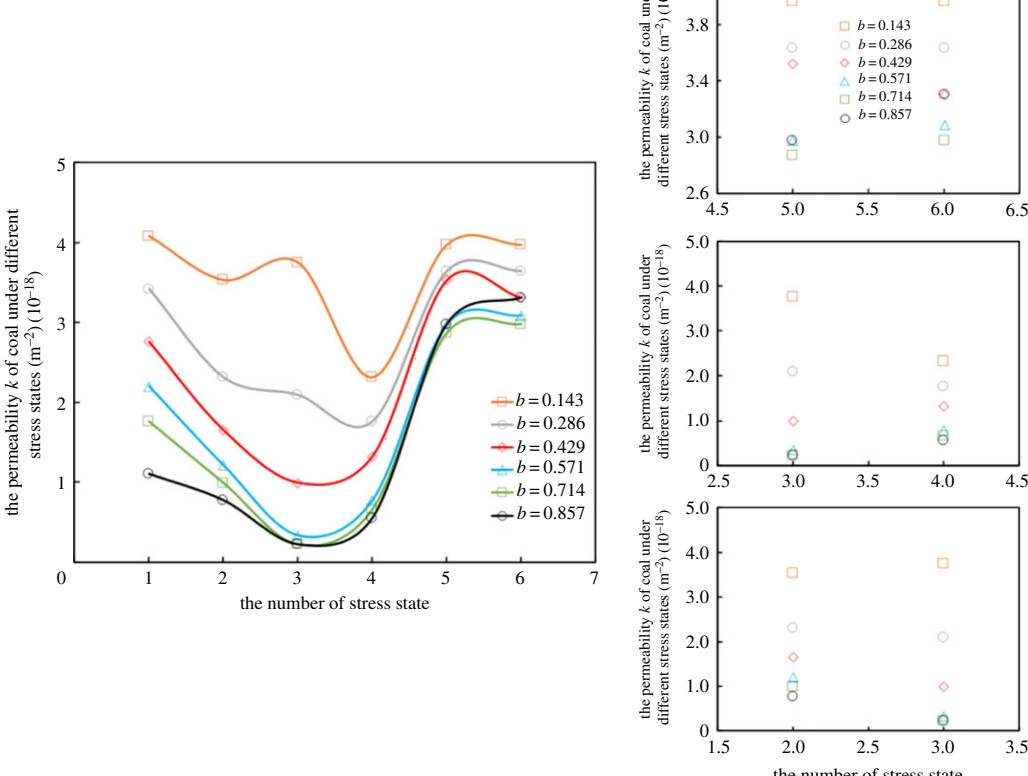

**Figure 14.** Variation of permeability $k$ under six kinds of stress states.

$(\sigma_Y, \sigma_Z) = (\sigma_{\|2}, \sigma_{\|1})$ on $k$ and $\varepsilon$ was uncertain in the above three sets of comparisons. The reasons were as follows: (1) when $\sigma_\perp$ was 15 MPa, the difference in $k$ was relatively small; (2) $\sigma_\|$ was perpendicular to the fluid flow direction ($Y$-direction). When $\sigma_\|$ was larger in this direction, the coal was compressed while extending the fluid flow path. Therefore, $k$ was lower under the combined effect of the above two conditions. In addition, when the stress arrangements were 1[#] and 5[#], 2[#] and 6[#], 3[#] and 5[#], 4[#] and 6[#], the permeability variations also showed that the $\sigma_\perp$ on permeability impact could not be ignored. It was not possible to simply assume raw coal as a transversely isotropic material. In addition, the trend of $\varepsilon_V$ was similar to that of $k$. For factors affecting the values and variations of permeability, bedding characteristics and gas flow directions, the principal stress distributions must be taken into account.

**Table 2.** The comparison of $k$ under different stress states. (1) " > ", " < ", " = " and " ≈ " refer to the size relationships of the first column and line elements; (2) the values of $k_5^\#$ and $k_6^\#$ are similar and assumed to be $k_5^\# \approx k_6^\#$.

| $k$ | $k_1^\#$ | $k_2^\#$ | $k_3^\#$ | $k_4^\#$ | $k_5^\#$ | $k_6^\#$ |
|---|---|---|---|---|---|---|
| $k_1^\#$ | = | > | > | > | < | < |
| $k_2^\#$ | < | = | > | > | < | < |
| $k_3^\#$ | < | < | = | < | < | < |
| $k_4^\#$ | < | < | > | = | < | < |
| $k_5^\#$ | > | > | > | > | = | <(≈) |
| $k_6^\#$ | > | > | > | > | >(≈) | = |

Note that $\sigma_\perp$ was the stress perpendicular to the bedding plane in MPa, and $\sigma_{\|1}$ and $\sigma_{\|2}$ are the stresses parallel to the bedding plane in MPa.

## 3.4. Analysis of lateral expansion coefficient during the loading process

By taking $Z$-direction loading as an example, each directional deformation was analysed. Owing to the Poisson effect, the influence of deformation in the three directions was mutual; that is, the strain result in one direction was the final magnitude incurred by the mutual function in all directions. This could be expressed as follows:

$$\Delta\varepsilon_{Z-\text{net}} = \Delta\varepsilon_Z - \nu_X \cdot \Delta\varepsilon_X - \nu_Y \cdot \Delta\varepsilon_Y \tag{3.6}$$

$$\text{and}\quad \Delta\sigma_Z = E_Z \cdot \Delta\varepsilon_{Z-\text{net}}. \tag{3.7}$$

Similarly, the following formulae could be obtained in $Y$ and $X$-direction loading:

$$\Delta\varepsilon_{Y-\text{net}} = \Delta\varepsilon_Y - \nu_X \cdot \Delta\varepsilon_X - \nu_Z \cdot \Delta\varepsilon_Z, \tag{3.8}$$

$$\Delta\sigma_Y = E_Y \cdot \Delta\varepsilon_{Y-\text{net}}, \tag{3.9}$$

$$\Delta\varepsilon_{X-\text{net}} = \Delta\varepsilon_X - \nu_Y \cdot \Delta\varepsilon_Y - \nu_Z \cdot \Delta\varepsilon_Z \tag{3.10}$$

$$\text{and}\quad \Delta\sigma_X = E_X \cdot \Delta\varepsilon_{X-\text{net}}. \tag{3.11}$$

where $\Delta\varepsilon_{j-\text{net}}$ is the strain variation caused by the external force in the $j$-direction ($j = X, Y, Z$); $\Delta\varepsilon_j$ is the ultimate strain variation under the interaction of three directions; $\nu_j$ and $E_j$ are lateral expansion coefficient and the elastic modulus in the $j$-direction, respectively, in MPa; $\Delta\sigma_j$ is the stress variable quantity in MPa.

It was assumed that the coal was always in a linear elastic state, and $\varepsilon_j$ was obtained by the generalized Hooke's Law:

$$\varepsilon_X = \frac{\sigma_X - \nu_x(\sigma_Y + \sigma_Z)}{E_X}, \tag{3.12a}$$

$$\varepsilon_Y = \frac{\sigma_Y - \nu_Y(\sigma_X + \sigma_Z)}{E_Y} \tag{3.12b}$$

$$\text{and}\quad \varepsilon_Z = \frac{\sigma_Z - \nu_Z(\sigma_X + \sigma_Y)}{E_Z}. \tag{3.12c}$$

Thus, the following formula was obtained:

$$\Delta\sigma_X = [\sigma_X - \nu_x(\sigma_Y + \sigma_Z)](\Delta\varepsilon_X - \nu_Y\Delta\varepsilon_Y - \nu_Z\Delta\varepsilon_Z)/\varepsilon_X, \tag{3.13a}$$

$$\Delta\sigma_Y = [\sigma_Y - \nu_Y(\sigma_X + \sigma_Z)](\Delta\varepsilon_Y - \nu_X\Delta\varepsilon_X - \nu_Z\Delta\varepsilon_Z)/\varepsilon_Y \tag{3.13b}$$

and
$$\Delta\sigma_Z = [\sigma_Z - \nu_Z(\sigma_X + \sigma_Y)](\Delta\varepsilon_Z - \nu_Y\Delta\varepsilon_Y - \nu_X\Delta\varepsilon_X)/\varepsilon_Z, \tag{3.13c}$$

When loaded to 20, 25, 30, 35, 40, 45 and 50 MPa, respectively, equations ((3.13a)–(3.13c)) could be used in order to calculate $\nu_j$. The results are shown in figure 15.

$\nu$ also exhibited anisotropic properties, such as: (1) $\nu$ meets $\nu_\perp < \nu_\|$; (2) when fixing $\sigma_1$ and $\sigma_3$, $\nu$ increases with $\sigma_2$ loading; (3) the difference of $\nu_{\|1}$ and $\nu_{\|2}$ was small. Note that $\nu_\perp$ was the lateral expansion coefficient perpendicular to the bedding plane, and $\nu_\|$ was the lateral expansion coefficient parallel to the bedding plane.

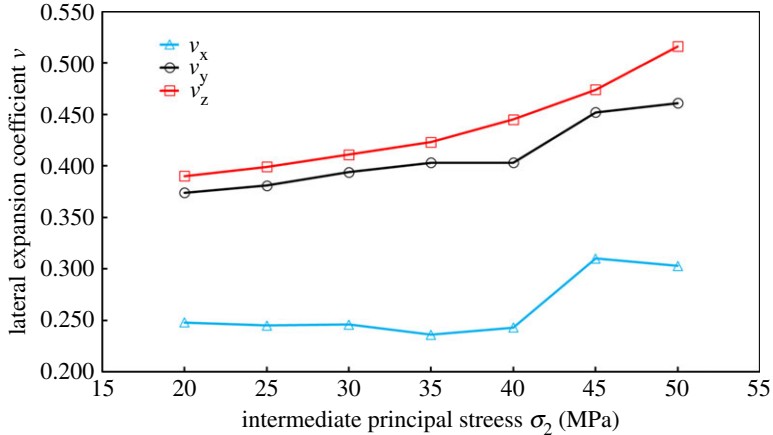

**Figure 15.** Variation of lateral expansion coefficient $\nu_j$ with intermediate principal stress $\sigma_2$ loading.

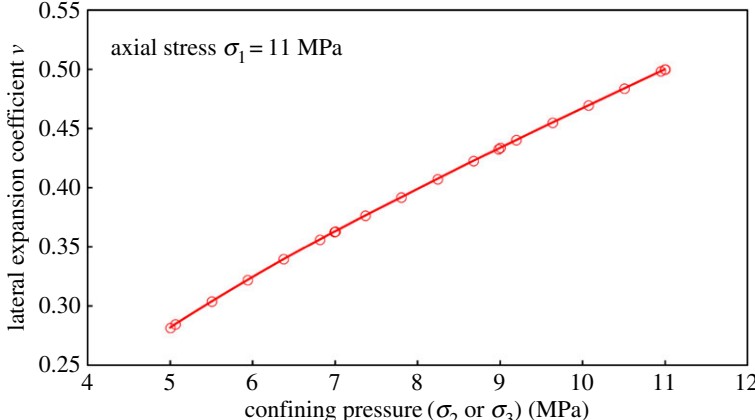

**Figure 16.** The variation of lateral expansion coefficient $\nu$ with confining pressure $\sigma_2$ under the condition of maintaining axial stress and increasing confining pressure in triaxial experiments.

By considering the Y-direction loading process as an example, since $\sigma_2$ was parallel to the bedding plane, tensile stress was generated around the bedding; that is, the deformation response to tensile stress was more sensitive. In summary, the performance was described by (1). When loaded from one stress level to a higher stress level, the lateral deformation degree increased with respect to axial deformation; therefore it was (2). This was also consistent with the increase of $\nu$ with confining pressure loading in conventional triaxial compression experiments, as shown in figure 16. Under the assumption that coals were a transversely isotropic material, $\nu$ was $\nu_Y = \nu_Z$. The author holds that the difference between $\nu_Y$ and $\nu_Z$ was due to coal characteristics, such as its internal bedding and disordered pores and cracks. However, for these special structures, there were some uncertainties in location, quantity and stress environment, and these may cause some differences to the Poisson effect in the Y-direction (or Z-direction). However, along the bedding direction, the strain response was similar under the action of force, and the resulting performance was (3).

The stress environment of the conventional triaxial compression experiment was $\sigma_1 = 11$ MPa, and $\sigma_3$ was loaded from 5 to 11 MPa.

## 4. Conclusion

This study carried out loading–unloading experiments on raw coals under true triaxial stress conditions. The purpose of the study was to investigate the deformation and permeability characteristics. The following conclusions were drawn:

(1) The variation of $k$ was determined by $\varepsilon_V$, stress, and loading–unloading paths. When $\sigma_2$ was parallel to beddings of the coal, the coal was most likely to dilate until it reached the yield situation for this stress environment. Additionally, $k$ was more likely to begin increasing after falling to the lowest level with $\sigma_2$ loading, even if it had been in a compression state.

(2) In comparison to coal skeleton, the beddings were easier to deform. The strain variation was largest of coal saturating $CO_2$ under applied stress and keeping gas pressure unchanged, regardless of loading or unloading in the direction vertical to bedding. In each direction, the variation of $\varepsilon_V$ was determined by $\varepsilon$, it was not subjected to the control of the initial compression degree, and belonged to process variation. Both of them were important factors affecting $k$, rather than the determinants.

(3) The variations of $(\varepsilon_1 - \varepsilon_3)$ and $\Delta k$ with $\sigma_2$ loading or unloading were consistent. The strain difference indicated the degree of expansion or compression in each direction under the action of force. That is, both of them represented the degree of difficulty by which the coal specimens could be compressed at the initial stress points and the degree of deformation in the $\varepsilon_1$ and $\varepsilon_3$ direction during the experimental process.

(4) In the same true triaxial stress state, $k$ was smaller when a larger force perpendicular to the bedding plane was applied. While keeping $\sigma_X = \sigma_\perp$ unchanged and $\sigma_{\parallel 1} > \sigma_{\parallel 2}$, the effects of $(\sigma_Y, \sigma_Z) = (\sigma_{\parallel 1}, \sigma_{\parallel 2})$, and $(\sigma_Y, \sigma_Z) = (\sigma_{\parallel 2}, \sigma_{\parallel 1})$ on $k$ and $\varepsilon$ were different. Additionally, the larger was the $\sigma_\parallel$ perpendicular to the fluid flow direction, the smaller was $k$.

(5) $\nu$ also exhibited anisotropic properties in three directions, such as $\nu$ complying to the following rules: (1) $\nu_\perp < \nu_\parallel$; (2) when $\sigma_1$ and $\sigma_3$ were fixed, $\nu$ increased as $\sigma_2$ increased; (3) $\nu_{\parallel 1}$ and $\nu_{\parallel 2}$ showed a certain difference.

Data accessibility. The datasets supporting this article have been uploaded as part of the electronic supplementary material.

Authors' contributions. M.L. and C.L. conceived and designed the experiments; J.D. and Z.S. performed the experiments; C.L. and J.D. analysed the data; and J.D. wrote the paper.

Competing interests. We declare we have no competing interests.

Funding. This research was funded by the Research Fund of State Key Laboratory for GeoMechanics and Deep Underground Engineering, CUMT (SKLGDUEK1809).

Acknowledgements. We appreciate the support of Dr Guansen Cao from Chongqing University.

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
