## [Reviewer comments · Royal Society Open Science]

Review History

RSOS-181438.R0 (Original submission)

Review form: Reviewer 1

Is the manuscript scientifically sound in its present form?

Yes

Are the interpretations and conclusions justified by the results?

Yes

Is the language acceptable?

Yes

Is it clear how to access all supporting data?

Not Applicable

Do you have any ethical concerns with this paper?

No

Have you any concerns about statistical analyses in this paper?

No

Recommendation?

Accept with minor revision (please list in comments)

Comments to the Author(s)

1. Why has the manuscript been uploaded twice?
2. Throughout the manuscript, no detailed information was provided regarding the experimental of strains and the permeability. Simply, they were mentioned rather than any proper information. Authors are advised to add detailed information.
3. No chemical analysis of the studied coal was included in the manuscript.
4. For the measurement of permeability, CO₂ was utilized. Authors might be aware of the swelling nature of coal during the interaction with the carbon dioxide. So, while measuring the permeability was the swelling factor was considered.
5. What can be the possible cause for the increase in the permeability of the coal after a certain value of the intermediate stress?
6. Can the PR of the coal be up to 0.5? Please verify with the calculations.

Review form: Reviewer 2 (Dengke Wang)

Is the manuscript scientifically sound in its present form?

Yes

Are the interpretations and conclusions justified by the results?

Yes

Is the language acceptable?

Yes

Is it clear how to access all supporting data?

Yes

Do you have any ethical concerns with this paper?

No

Have you any concerns about statistical analyses in this paper?

No

Recommendation?

Accept with minor revision (please list in comments)

Comments to the Author(s)

The manuscript presented an interesting investigation on permeability and deformation properties of coals containing bedding planes under true triaxial stress conditions. The manuscript seems organized well. The approach taken and result are interesting and useful for

potential readers. Specific comments for revision are as follows,

- (1) In abstract, the language is confused and the logic is not clear, so it needs to be completely revised.
- (2) Why not consider the effective stress effect?
- (3) How did you make sure that flow regime was dominated by Darcy flow during experiment?
- (4) For the rigid loading, the end effect inevitably exist, which can affect the deformation measurement results of the specimen. What methods did you use to reduce the end friction effect?
- (5) How did you measure sample deformation in each direction, especially for volumetric strain?
- (6) Please provide in-situ stress conditions in the operation field where you took this specimen from?
- (7) Explain why the gas pressure in this paper is 3 MPa?
- (8) I recommended the author delete the figure about ($\epsilon_1 - \epsilon_2$) and ($\epsilon_2 - \epsilon_3$) in Fig. 12.
- (9) Please indicate the beddings in the coal specimen.

Review form: Reviewer 3

Is the manuscript scientifically sound in its present form?

No

Are the interpretations and conclusions justified by the results?

No

Is the language acceptable?

Yes

Is it clear how to access all supporting data?

Not Applicable

Do you have any ethical concerns with this paper?

No

Have you any concerns about statistical analyses in this paper?

No

Recommendation?

Major revision is needed (please make suggestions in comments)

Comments to the Author(s)

It is very confused regarding the primary objective of the study. Do the authors wanna figure out the role of bedding planes in fluid flow in coal reservoirs? The main problem with the study is that it lacks novelty. Some detailed comments are as follows:

1. Line 39: this is a completely incorrect statement. Lots of studies have proved that bedding planes of coal have little influence on coal permeability.
2. In Introduction, more than 90% of the content is about sandstone and shale. The goal of the study is to investigate the mechanical behavior and coal permeability, so the authors should pay their attention to coal. Considering the significant difference in reservoir structure between coal and other reservoir rocks, the studies on other rocks and the corresponding findings cannot be simply embezzled to coal.
3. How many specimens were prepared and tested?

4. Section 2.3, the experimental process is far from clear. What is the purpose for each step? More importantly, what kind of gas was used for testing? Different conclusions can be drawn when using different test gases.
5. Line 159-160: it is impossible to get the conclusion from Fig. 5.
6. In Conclusion: "the strain variation was largest, regardless of loading or unloading in the direction vertical to bedding". This conclusion is incorrect when it comes to coal if using non-sorptive gas as the test gas.
7. No new findings are reported in this study.

Decision letter (RSOS-181438.R0)

20-Nov-2018

Dear Dr Liu,

The editors assigned to your paper ("Influence of Principal Stress Effect on Deformation and Permeability of Coal Containing beddings under True Triaxial Stress Conditions") have now received comments from reviewers. We would like you to revise your paper in accordance with the referee and Associate Editor suggestions which can be found below (not including confidential reports to the Editor). Please note this decision does not guarantee eventual acceptance.

Please submit a copy of your revised paper before 13-Dec-2018. Please note that the revision deadline will expire at 00.00am on this date. If we do not hear from you within this time then it will be assumed that the paper has been withdrawn. In exceptional circumstances, extensions may be possible if agreed with the Editorial Office in advance. We do not allow multiple rounds of revision so we urge you to make every effort to fully address all of the comments at this stage. If deemed necessary by the Editors, your manuscript will be sent back to one or more of the original reviewers for assessment. If the original reviewers are not available, we may invite new reviewers.

If your study uses humans or animals please include details of the ethical approval received, including the name of the committee that granted approval. For human studies please also detail

whether informed consent was obtained. For field studies on animals please include details of all permissions, licences and/or approvals granted to carry out the fieldwork.

- Data accessibility

If you wish to submit your supporting data or code to Dryad (<http://datadryad.org/>), or modify your current submission to dryad, please use the following link:
<http://datadryad.org/submit?journalID=RSOS&manu=RSOS-181438>

- Competing interests

- Authors' contributions

- Acknowledgements

- Funding statement

Please note that Royal Society Open Science charge article processing charges for all new submissions that are accepted for publication. Charges will also apply to papers transferred to Royal Society Open Science from other Royal Society Publishing journals, as well as papers submitted as part of our collaboration with the Royal Society of Chemistry (<http://rsos.royalsocietypublishing.org/chemistry>). If your manuscript is newly submitted and subsequently accepted for publication, you will be asked to pay the article processing charge, unless you request a waiver and this is approved by Royal Society Publishing. You can find out

more about the charges at <http://rsos.royalsocietypublishing.org/page/charges>. Should you have any queries, please contact openscience@royalsociety.org.

on behalf of Professor R. Kerry Rowe (Subject Editor)
openscience@royalsociety.org

Associate Editor's comments:

Thank you for the submission. We have now received three reports on your paper. Each reviewer has matters they would like to see addressed before the paper may be considered for publication. In particular, you will note one of the reviewers questions the degree of novelty your work represents. Royal Society Open Science requires manuscripts to represent a meaningful contribution to the literature -- at present, it is not clear how your paper fulfills this criterion. Please address the concerns of the reviews, and this last point in particular. If your revision does not persuade the referee that your paper represents an advance on existing knowledge, we regret it will not be possible to consider the paper further for publication. Good luck and we look forward to receiving your response.

Comments to Author:

Reviewers' Comments to Author:

Reviewer: 1

Comments to the Author(s)

1. Why has the manuscript been uploaded twice?
2. Throughout the manuscript, no detailed information was provided regarding the experimental of strains and the permeability. Simply, they were mentioned rather than any proper information. Authors are advised to add detailed information.
3. No chemical analysis of the studied coal was included in the manuscript.
4. For the measurement of permeability, CO₂ was utilized. Authors might be aware of the swelling nature of coal during the interaction with the carbon dioxide. So, while measuring the permeability was the swelling factor was considered.
5. What can be the possible cause for the increase in the permeability of the coal after a certain value of the intermediate stress?
6. Can the PR of the coal be up to 0.5? Please verify with the calculations.

Reviewer: 2

Comments to the Author(s)

The manuscript presented an interesting investigation on permeability and deformation properties of coals containing bedding planes under true triaxial stress conditions. The manuscript seems organized well. The approach taken and result are interesting and useful for potential readers. Specific comments for revision are as follows,

- (1) In abstract, the language is confused and the logic is not clear, so it needs to be completely revised.
- (2) Why not consider the effective stress effect?
- (3) How did you make sure that flow regime was dominated by Darcy flow during experiment?
- (4) For the rigid loading, the end effect inevitably exist, which can affect the deformation measurement results of the specimen. What methods did you use to reduce the end friction effect?
- (5) How did you measure sample deformation in each direction, especially for volumetric strain?
- (6) Please provide in-situ stress conditions in the operation field where you took this specimen from?
- (7) Explain why the gas pressure in this paper is 3 MPa?
- (8) I recommended the author delete the figure about ($\epsilon_1 - \epsilon_2$) and ($\epsilon_2 - \epsilon_3$) in Fig. 12.
- (9) Please indicate the beddings in the coal specimen.

Reviewer: 3

Comments to the Author(s)

It is very confused regarding the primary objective of the study. Do the authors wanna figure out the role of bedding planes in fluid flow in coal reservoirs? The main problem with the study is that it lacks novelty. Some detailed comments are as follows:

1. Line 39: this is a completely incorrect statement. Lots of studies have proved that bedding planes of coal have little influence on coal permeability.
2. In Introduction, more than 90% of the content is about sandstone and shale. The goal of the study is to investigate the mechanical behavior and coal permeability, so the authors should pay their attention to coal. Considering the significant difference in reservoir structure between coal and other reservoir rocks, the studies on other rocks and the corresponding findings cannot be simply embezzled to coal.
3. How many specimens were prepared and tested?
4. Section 2.3, the experimental process is far from clear. What is the purpose for each step? More importantly, what kind of gas was used for testing? Different conclusions can be drawn when using different test gases.
5. Line 159-160: it is impossible to get the conclusion from Fig. 5.
6. In Conclusion: "the strain variation was largest, regardless of loading or unloading in the direction vertical to bedding". This conclusion is incorrect when it comes to coal if using non-sorptive gas as the test gas.
7. No new findings are reported in this study.

Author's Response to Decision Letter for (RSOS-181438.R0)

See Appendix A.

RSOS-181438.R1 (Revision)

Review form: Reviewer 1

Is the manuscript scientifically sound in its present form?

Yes

Are the interpretations and conclusions justified by the results?

Yes

Is the language acceptable?

Yes

Is it clear how to access all supporting data?

Yes

Do you have any ethical concerns with this paper?

No

Have you any concerns about statistical analyses in this paper?

No

Recommendation?

Accept with minor revision (please list in comments)

Comments to the Author(s)

Formatting error: page 190-194

In all the figures, some more information can be added to the caption mentioning what does in illustrates.

Review form: Reviewer 2 (Dengke Wang)

Is the manuscript scientifically sound in its present form?

Yes

Are the interpretations and conclusions justified by the results?

Yes

Is the language acceptable?

Yes

Is it clear how to access all supporting data?

Yes

Do you have any ethical concerns with this paper?

No

Have you any concerns about statistical analyses in this paper?

No

Recommendation?

Accept as is

Comments to the Author(s)

The responds of authors are reasonable and the revised version of the paper can be also acceptable.

Review form: Reviewer 3**Is the manuscript scientifically sound in its present form?**

Yes

Are the interpretations and conclusions justified by the results?

Yes

Is the language acceptable?

Yes

Is it clear how to access all supporting data?

Not Applicable

Do you have any ethical concerns with this paper?

No

Have you any concerns about statistical analyses in this paper?

No

Recommendation?

Accept as is

Comments to the Author(s)

The authors have addressed all my concerns, and the manuscript is currently suggested for publication.

Decision letter (RSOS-181438.R1)

15-Jan-2019

Dear Dr Liu:

On behalf of the Editors, I am pleased to inform you that your Manuscript RSOS-181438.R1 entitled "Influence of Principal Stress Effect on Deformation and Permeability of Coal Containing beddings under True Triaxial Stress Conditions" has been accepted for publication in Royal Society Open Science subject to minor revision in accordance with the referee suggestions. Please find the referees' comments at the end of this email.

The reviewers and Subject Editor have recommended publication, but also suggest some minor

revisions to your manuscript. Therefore, I invite you to respond to the comments and revise your manuscript.

- Ethics statement

- Data accessibility

If you wish to submit your supporting data or code to Dryad (<http://datadryad.org/>), or modify your current submission to dryad, please use the following link:
<http://datadryad.org/submit?journalID=RSOS&manu=RSOS-181438.R1>

- Competing interests

- Authors' contributions

- Acknowledgements

- Funding statement

Please note that we cannot publish your manuscript without these end statements included. We

have included a screenshot example of the end statements for reference. If you feel that a given heading is not relevant to your paper, please nevertheless include the heading and explicitly state that it is not relevant to your work.

Because the schedule for publication is very tight, it is a condition of publication that you submit the revised version of your manuscript before 24-Jan-2019. Please note that the revision deadline will expire at 00.00am on this date. If you do not think you will be able to meet this date please let me know immediately.

on behalf of Professor R. Kerry Rowe (Subject Editor)
openscience@royalsociety.org

Associate Editor Comments to Author:

Thank you for so successfully addressing the referees' concerns. We'd like you to review the captions per the referee's remaining comment, but otherwise, good job!

Reviewer comments to Author:

Reviewer: 2

Comments to the Author(s)

The responds of authors are reasonable and the revised version of the paper can be also acceptable.

Reviewer: 1

Comments to the Author(s)

Formatting error: page 190-194

In all the figures, some more information can be added to the caption mentioning what does in illustrates.

Reviewer: 3

Comments to the Author(s)

The authors have addressed all my concerns, and the manuscript is currently suggested for publication.

Author's Response to Decision Letter for (RSOS-181438.R1)

See Appendix B.

Decision letter (RSOS-181438.R2)

17-Jan-2019

Dear Dr Liu,

I am pleased to inform you that your manuscript entitled "Influence of Principal Stress Effect on Deformation and Permeability of Coal Containing beddings under True Triaxial Stress Conditions" is now accepted for publication in Royal Society Open Science.

You can expect to receive a proof of your article in the near future. Please contact the editorial office (openscience_proofs@royalsociety.org and openscience@royalsociety.org) to let us know if

you are likely to be away from e-mail contact. Due to rapid publication and an extremely tight schedule, if comments are not received, your paper may experience a delay in publication.

on behalf of Professor R. Kerry Rowe (Subject Editor)
openscience@royalsociety.org

Appendix A

Responses to reviewers and editors

Manuscript Number: RSOS-181438

Paper Title: Influence of Principal Stress Effect on Deformation and Permeability of Coal
Containing beddings under True Triaxial Stress Conditions

Authors: Jiahui Dai, Chao Liu, Minghui Li and Zhenlong Song

Dear editors and reviewers,

Special thanks to the anonymous reviewers for their careful reviews and valuable comments. We have carefully checked these comments and made necessary corrections and modifications. We tried our best to improve the manuscript.

Improvements and clarifications have been made in the current manuscript. These changes are marked by **red words**. English grammar errors and inaccurate expressions are modified and rephrased.

We appreciate for the editors and reviewers' warm work earnestly, and hope that the manuscript after the corrections can satisfy the requirements for publication. Once again, thank you very much for your comments and suggestions.

Point-by-point Responses to the reviewers' comments

Reviewer #1:

Comment 1: Why has the manuscript been uploaded twice?

Response:

Thanks for the comment. We are very sorry for it. Since we are the first to the Journal: Royal Society Open Science, we are not familiar with the submission system. It's also related to our carelessness. If we have the opportunity to submit new manuscripts to the

journal, we will strictly comply with the requirements of the submission system and avoid such mistakes.

Comment 2: Throughout the manuscript, no detailed information was provided regarding the experimental of strains and the permeability. Simply, they were mentioned rather than any proper information. Authors are advised to add detailed information.

Response:

Thanks for the comment. The coals extracted from the well were processed to get the cube specimens, and then we started the experiment. The experimental steps are as follows:

(1) Place the specimen on the loading plate, and then put on a heat-shrinkable tube, which should cover the sealing washer of the loading plate (Fig. R1). The heat-shrinkable tube is tightly attached to the surface of the specimen with a hot air gun. Then the heat-shrinkable tube is tightly fastened at the sealing washer to prevent air leakage during the experiment.

(2) The specimen is fed into the loading cell, and the LVDT is installed and connected. The force of the true triaxial control system is cleared and the loading plate is moved slowly towards the specimen by displacement control. Meanwhile, the indicator changes of the LVDT displayed on the system are observed. When the system shows that the force of the six loading plates is 0.5 kN to 2.5 kN (indicating that the loading plate has been in contact with the specimen) and the LVDT is within -1 mm ~ +1mm (the range of LVDT is -5 mm ~ +5mm), the installation of the LVDT is feasible. Otherwise, the LVDT will be readjusted (Measurements of the specimen's deformation are made independently along

each principal stress axis using the linear variable displacement transducers as shown in Fig. R2).

(3) Seal the loading cell and fill it with hydraulic oil. When it is full, close the intake valve. The hydraulic oil was pressurized to 5 MPa by the servo valve (according to the principle of force and reaction force, the stress designed by the experiment is correspondingly reduced by 5 MPa) and the oil pressure should be greater than the gas pressure in the experiment to prevent the heat-shrinkable tube from bursting. Then enter the gas and adjust the pressure valve to 3 MPa.

(4) The experiment is carried out according to the designed stress path. When the stress reaches each stress point, the change of the LVDT and the flow rate of the flowmeter are observed. The next step can only be taken after the displacement and flow are stable.

Fig. R1. Multi-functional loading platen.

Fig. R2. Loading cell.

In the experiment, what we are most concerned about is the change of LVDT. The strain can be obtained by the displacement data recorded by the data acquisition system, which is the basis of the mechanical analysis.

We applied steady state method to measure permeability. For steady state gas flow measurements, the inlet pressure was kept constant while the gas on the outlet side was at atmospheric pressure ($P_{out}=0.1$ MPa). The inlet and outlet pressure data were recorded in short intervals (1.2 s for the control and data acquisition system of the multifunctional true triaxial geophysical apparatus) using the pressure transducers. The gas flow rate at the outlet side was monitored using a mass flow meter with the precision of 1mL/min. To start a steady state measurement, the inlet pressure was adjusted to a desired pressure (it is 3 MPa in the paper) using the pressure regulator connected to the gas cylinder. Once the steady state flow conditions were reached, the flow rates were recorded and a higher stress

was applied in accordance with the stress path. Then the permeability was calculated by Darcy's law.

Comment 3: No chemical analysis of the studied coal was included in the manuscript.

Response:

Thanks for the comment. Hu et al. (2018) concluded that in situ stresses have a direct impact on the potential for fracture shear failure, the evolution of stress-dependent aperture and the corresponding evolution of permeability [1]. Therefore, in the paper, the purpose of the research is to investigate the deformation mechanism of the coal and gas migration in various directions under the effect of bedding. For studied coal, we only conducted proximate analysis as shown in Table R1.

Table R1. Proximate analysis of studied coal.

M/%	A/%	V/%	Fc/%
1.07	24.37	9.7	65.57

where M is the moisture, A is the coal ash, V is the coal volatile, Fc is fixed carbon.

Reference:

[1] Hu Y, Gan Q, Hurst A, Elsworth D. Evolution of permeability in sand injectite systems. *International Journal of Rock Mechanics and Mining Sciences*, 2018, 106, 176–189. (doi:10.1016/j.ijrmms.2018.04.018)

Comment 4: For the measurement of permeability, CO₂ was utilized. Authors might

be aware of the swelling nature of coal during the interaction with the carbon dioxide. So, while measuring the permeability was the swelling factor was considered.

Response:

Thanks for the comment. It is true that CO₂ adsorption will cause the coal matrix swelling. Hence, the total strain and permeability of the coal are affected by CO₂ adsorption. However, it is very difficult to figure out the actual effects of CO₂ adsorption on the deformation and permeability quantitatively. Besides, in our experiments, the gas pressure was constant (3 MPa). During the whole permeability measurements, the inlet pressure kept at 3 MPa and the outlet pressure kept at 0.1 MPa. As the gas pressure kept constant, the effects of CO₂ adsorption on the deformation and permeability kept constant as well. Therefore, the measured strain and permeability were under comprehensive conditions, which can reflect the real field situations. Thank you again for the comment.

Comment 5: What can be the possible cause for the increase in the permeability of the coal after a certain value of the intermediate stress?

Response:

Thanks for the comment. Rock strength firstly increases and subsequently decreases with the increase of intermediate principal stress [2]. When the intermediate principal stress increases to the critical value, the resistance to deformation of coals is weakened, which makes the pores and cracks in coal body tend to open from the original closed state, and finally show an increase in permeability.

Reference:

[2] Pan P Z, Feng X T, Hudson J A. The influence of the intermediate principal stress on rock failure behaviour: A numerical study. *Engineering Geology*, 2012, 124:109-118. (doi.org/10.1016/j.enggeo.2011.10.008)

Comment 6: Can the PR of the coal be up to 0.5? Please verify with the calculations.

Response:

Thanks for the comment. In this paper, we conducted triaxial compression experiments, not uniaxial compression. Therefore, it is more appropriate to refer to μ as the lateral expansion coefficient. Substituting the true triaxial experimental data into Eqs. (13) and (14) of original manuscript, we obtain the variation of the lateral expansion coefficient versus the intermediate principal stress. Under conventional triaxial compression experiments, Figure (18) in original manuscript is derived from the following Eqs. :

$$\mu = \frac{B\sigma_1 - \sigma_3}{\sigma_3 - 2B - 1 - \sigma_1} \quad (\text{R1})$$

$$B = \varepsilon_3 / \varepsilon_1 \quad (\text{R2})$$

The Poisson's ratio in the full text is changed to the lateral expansion coefficient.

Reviewer #2:

Comment 1: In abstract, the language is confused and the logic is not clear, so it needs to be completely revised.

Response:

Thanks for the comment. According to the reviewer's suggestion, the abstract is modified as follows: In-situ stress is generally an anisotropic/true triaxial stress ($\sigma_1 > \sigma_2 > \sigma_3$). Bedding weaken the continuity and integrity of coal. It is critical to understand the mechanical behavior and gas migration of coal under true triaxial stress conditions. We performed experiments of cubic coal samples to investigate the permeability evolution and mechanical behavior of coal under true triaxial stress conditions by using newly developed true triaxial geophysical (TTG) apparatus. We analyzed the effect of principal stresses on deformation and permeability characteristics of coal containing bedding planes. The results show that volumetric strain, stress states and bedding directions determine the permeability comprehensively. The variable quantity of strain was the largest in the direction normal to the bedding plane. The expansion or compression degree were characterized by the difference between the major and minor principal strain ($\epsilon_1 - \epsilon_3$). Essentially, this represents the difficulty degree with regard to coal being compressed at the initial stress state, and the deformation degree in ϵ_1 and ϵ_3 direction. The variation of ($\epsilon_1 - \epsilon_3$) was consistent to that of permeability. Under an identical true triaxial stress condition, permeability was smaller when larger stress was applied in the direction normal to the bedding plane. Additionally, stress level in the direction parallel to the bedding planes and the directions between stresses in the direction paralleling to the bedding planes and the flow direction also affect the permeability and strain. By solving lateral expansion coefficient, coal also exhibited anisotropic properties.

Comment 2: Why not consider the effective stress effect?

Response:

Thank you for this comment. In our experiments, the gas pressure was constant (3 MPa). During the whole permeability measurements, the inlet pressure kept at 3 MPa and the outlet pressure kept at 0.1 MPa. As the gas pressure kept constant, the variations of total stress and effective stress were the same. The overall trends of the curves plotting in terms of total stress and effective stress are the same. Therefore, we used total stress in the manuscript.

Comment 3: How did you make sure that flow regime was dominated by Darcy flow during experiment?

Response:

Thanks for the comment. Fracture permeability is generally governed by Darcy's law [1, 2, 3]. Zhang and Wang also believed that the Darcy flow works within macropores (pore width > 50 nm) and fractures [4]. Wu et al. concluded that the flow rate, q , of $8.33 \times 10^{-7} \text{ m}^3/\text{s}$ is sufficiently low to achieve the steady-state flow [5]. The magnitude of flow rate under true triaxial conditions we measured is also between 10^{-6} and 10^{-7} , which indicates that the gas flow through the coal body is in steady-state. The Darcy's flow equation has been widely used in the field to estimate fluid permeability through porous media under steady-state flow conditions [6].

In addition, we have measured the coal flow rate, q , under different stresses and gas pressures. As shown in Fig. R3, the flow rate and gas pressure are linearly fitted with a

correlation coefficient, R^2 of 0.9981 and 0.9987. It also proves that we use the rationality of Darcy's law equation.

Fig. R3. Flow rate versus gas pressure under specific stress conditions.

Reference:

- [1] Yin G, Li M, Wang J G, Xu J, Li W. Mechanical behavior and permeability evolution of gas infiltrated coals during protective layer mining. *International Journal of Rock Mechanics & Mining Sciences*, 2015; 80:292-301. (doi:10.1016/j.ijrmms.2015.08.022)
- [2] Liu Y, Li M, Yin G, Zhang D, Deng B. Permeability evolution of anthracite coal considering true triaxial stress conditions and structural anisotropy. *Journal of Natural Gas Science & Engineering*, 2018, 52:492-506. (doi:10.1016/j.jngse.2018.02.014)
- [3] Perera M S A, Ranjith P G, Choi S K, Airey D. The effects of sub-critical and super-critical carbon dioxide adsorption-induced coal matrix swelling on the permeability of naturally fractured black coal. *Energy*, 2011, 36(11):6442-6450. (doi:10.1016/j.energy.2011.09.023)
- [4] Zhang W & Wang Q. Permeability anisotropy and gas slippage of shales from the Sichuan Basin in South China. *International Journal of Coal Geology*, 2018,194, 22–32. (doi:10.1016/j.coal.2018.05.004)

[5] Wu W, Reece J S, Gensterblum Y, Zoback M D. Permeability Evolution of Slowly Slipping Faults in Shale Reservoirs. *Geophysical Research Letters*, 2017, 44(22), 11368–11375. (doi:10.1002/2017gl075506)

[6] De Silva G P D, Ranjith P G, Perera M S A, Chen B. Effect of bedding planes, their orientation and clay depositions on effective re-injection of produced brine into clay rich deep sandstone formations: Implications for deep earth energy extraction. *Applied Energy*, 2016, 161:24-40. (doi:10.1016/j.apenergy.2015.09.079)

Comment 4: For the rigid loading, the end effect inevitably exist, which can affect the deformation measurement results of the specimen. What methods did you use to reduce the end friction effect?

Response:

Thanks for the comment. The mismatch in the elastic parameters (Young's modulus and Poisson's ratio) between the coal specimen and the metal platens of Multi-functional true triaxial geophysical apparatus produces an interface friction when loaded in six directions, which results in a non-uniform stress distribution at the end of the specimen. Based on the rigid loading mode, this situation is inevitable, and we call it the end friction effect [7]. Lubrication is an effective method to decrease the friction between metal loading-platens and specimen faces by applying anti-friction agent. In this paper, the first step is to wrap the coal specimen with a heat-shrinkable tube, and then use a heat air gun to tighten the heat-shrinkable tube to the coal specimen. Finally, daub lithium grease evenly

on four horizontal loading platens. Due to ventilation for permeability experiment, no lubricant is applied to the vertical surface.

Reference:

[7] Feng, X.-T., Zhang, X., Yang, C., Kong, R., Liu, X., & Peng, S. Evaluation and reduction of the end friction effect in true triaxial tests on hard rocks. *International Journal of Rock Mechanics & Mining Sciences*, 2017, 97:144-148. (doi:10.1016/j.ijrmms.2017.04.002)

Comment 5: How did you measure sample deformation in each direction, especially for volumetric strain?

Response:

Thanks for the comment. The coals extracted from the well were processed to get the cube specimens, and then we started the experiment. The experimental steps are as follows:

(1) Place the specimen on the loading plate, and then put on a heat-shrinkable tube, which should cover the sealing washer of the loading plate (Fig. R1). The heat-shrinkable tube is tightly attached to the surface of the specimen with a hot air gun. Then the heat-shrinkable tube is tightly fastened at the sealing washer to prevent air leakage during the experiment.

(2) The specimen is fed into the loading cell, and the LVDT is installed and connected. The force of the true triaxial control system is cleared and the loading plate is moved slowly towards the specimen by displacement control. Meanwhile, the indicator changes of the LVDT displayed on the system are observed. When the system shows that the force of the six loading plates is 0.5 kN to 2.5 kN (indicating that the loading plate has been in

contact with the specimen) and the LVDT is within -1 mm ~ +1mm (the range of LVDT is -5 mm ~ +5mm), the installation of the LVDT is feasible. Otherwise, the LVDT will be readjusted (Measurements of the specimen's deformation are made independently along each principal stress axis using the linear variable displacement transducers as shown in Fig. R2).

(3) Seal the loading cell and fill it with hydraulic oil. When it is full, close the intake valve. The hydraulic oil was pressurized to 5 MPa by the servo valve (according to the principle of force and reaction force, the stress designed by the experiment is correspondingly reduced by 5 MPa) and the oil pressure should be greater than the gas pressure in the experiment to prevent the heat-shrinkable tube from bursting. Then enter the gas and adjust the pressure valve to 3 MPa.

(4) The experiment is carried out according to the designed stress path. When the stress reaches each stress point, the change of the LVDT and the flow rate of the flowmeter are observed. The next step can only be taken after the displacement and flow are stable.

Finally, the volumetric strain ε_V is obtained according to the volumetric strain expression $\varepsilon_V = \varepsilon_1 + \varepsilon_2 + \varepsilon_3$.

Comment 6: Please provide in-situ stress conditions in the operation field where you took this specimen from?

Response:

Thanks for the comment. The principal stress of the fully mechanized working face 2461 of outburst coal seam C₁ is shown in Table R2.

Table R2 In-situ stress of the fully mechanized working face 2461.

In-situ stress	Magnitude/ MPa	Dip angle/°	Azimuthal angle/°
Maximum principal stress	26.6	-6.7	90.1
Intermediate principal stress	19.4	-16.8	358.1
Minimum principal stress	8.5	-71.8	201.1

Comment 7: Explain why the gas pressure in this paper is 3 MPa?

Response:

Thanks for the comment. Based on the gas pressure of coal seam between 1.57 and 3.2 MPa, the experimental gas pressure was set to 3 MPa.

Comment 8: I recommended the author delete the figure about $(\epsilon_1 - \epsilon_2)$ and $(\epsilon_2 - \epsilon_3)$

in Fig. 12.

Response:

Thanks for the comment. According to the reviewer's suggestion, the Fig. 12 in original manuscript is modified as follows:

Fig. R4. The variation of $(\epsilon_1 - \epsilon_3)$ with σ_2 loading-unloading.

Comment 9: Please indicate the beddings in the coal specimen.

Response:

Thanks for the comment. The bedding in the coal specimen used in the experiment is shown Fig. R5.

Fig. R5. Coal specimens.

Reviewer #3:

Comment 1: Line 39: this is a completely incorrect statement. Lots of studies have proved that bedding planes of coal have little influence on coal permeability.

Response:

Thanks for the comment. Coal cleats are of two types: face cleats and butt cleat, which are often normal to the bedding plane and may be perpendicular to each other [1, 2, 3]. Fordsham and Gayer concluded that bedding plane slipping fractures are also tectonically induced and are widespread in coal seams, commonly associated with cleat [4]. It is well known that the coal permeability decrease significantly with cleat closure [5]. However, a lot of researches have been done on the effect of bedding on permeability and mechanical properties. The permeability varies extensively between different orientations, especially in the vertical and parallel directions of coal bedding [3, 6, 7]. Koenig et al. conducted permeability experiments on coals obtained from the Warrior Basin in the United States. The permeability ratio of different bedding directions reached a maximum of 17 : 1 [6]. Pomeroy et al. used water for the seepage test and found that when the confining pressure was perpendicular to the bedding plane, the permeability changed significantly [8]. Liu et al. and Li et al. carried out permeability tests under different bedding and joint conditions. They pointed out that the bedding and joint structure had an important influence on the permeability and deformation [7, 9]. Pan et al. took coal with different bedding directions as the research object, and studied the law of the permeability evolution during the loading process. It was concluded that drilling across bedding planes for gas extraction achieved better effects, at last reduce gas release and gas disasters [10]. Liang et al. conducted uniaxial compression tests on salt rock and coal specimens to obtain relevant basic mechanical parameters and strength characteristics of salt rock and coal in different bedding directions [11]. Therefore, the understanding of the contribution of bedding on seepage and deformation properties of coals containing beddings is both important and

necessary.

Reference:

- [1] Pan Z, & Connell L D. Modelling permeability for coal reservoirs: A review of analytical models and testing data. *International Journal of Coal Geology*, 2012, 92:1-44. (doi:10.1016/j.coal.2011.12.009)
- [2] Chen D, Pan Z, Ye Z, Hou B., Wang D, Yuan L. A unified permeability and effective stress relationship for porous and fractured reservoir rocks. *Journal of Natural Gas Science & Engineering*, 2016, 29:401-412. (doi:10.1016/j.jngse.2016.01.034)
- [3] Laubach S , Marrett R , Olson J, Scott A. Characteristics and origins of coal cleat: A review. *International Journal of Coal Geology*, 1998, 35(1–4):175-207. (doi:10.1016/s0166-5162(97)00012-8)
- [4] Frodsham K & Gayer R. The impact of tectonic deformation upon coal seams in the South Wales coalfield, UK. *International Journal of Coal Geology*, 1999, 38(3–4):297-332. (doi:10.1016/s0166-5162(98)00028-7)
- [5] Mckee C R & Bumb A C. Stress-dependent permeability and porosity of coal. *Spe Formation Evaluation*, 1988, 3(1):81-91. (doi:10.2118/12858-PA)
- [6] Koenig R A, Stubbs P B. Interference testing of a coalbed methane reservoir. *SPE Unconventional Gas Technology Symposium*, 1986. (doi:10.2118/15225-ms)
- [7] Wang S, Elsworth D, Liu J. Permeability evolution in fractured coal: The roles of fracture geometry and water-content. *International Journal of Coal Geology*, 2011, 87(1):13-25. (doi:10.1016/j.coal.2011.04.009)
- [8] Pomeroy C D & Robinson D J. The effect of applied stresses on the permeability of a

middle rank coal to water. *International Journal of Rock Mechanics & Mining Sciences & Geomechanics Abstracts*, 1967, 4(3):329-343. (doi:10.1016/0148-9062(67)90014-9)

[9] Li H, Shimada S, Zhang M. Anisotropy of gas permeability associated with cleat pattern in a coal seam of the Kushiro coalfield in Japan. *Environmental Geology*, 2004, 47(1):45-50. (doi:10.1007/s00254-004-1125-x)

[10] Pan R, Cheng Y, Yuan L, Yu M, Dong J. Effect of bedding structural diversity of coal on permeability evolution and gas disasters control with coal mining. *Natural Hazards*, 2014, 73(2):531-546. (doi:10.1007/s11069-014-1086-7)

[11] Liang W, Yang C, Zhao Y, Dusseault M B, Liu J. Experimental investigation of mechanical properties of bedded salt rock. *International Journal of Rock Mechanics & Mining Sciences*, 2007, 44(3):400-411. (doi:10.1016/j.ijrmms.2006.09.007)

Comment 2: In Introduction, more than 90% of the content is about sandstone and shale. The goal of the study is to investigate the mechanical behavior and coal permeability, so the authors should pay their attention to coal. Considering the significant difference in reservoir structure between coal and other reservoir rocks, the studies on other rocks and the corresponding findings cannot be simply embezzled to coal.

Response:

Thanks for the comment. We have added some literatures about the influence of bedding in coals on deformation and permeability evolutions, and deleted some literatures which are not relevant to the subject. The modified introduction are written in revised manuscript by red words.

Comment 3: How many specimens were prepared and tested?

Response:

Thank you for this comment. We performed the same experiments using two specimens. The results show similar deformation behaviors and permeability evolutions of the specimen.

Comment 4: Section 2.3, the experimental process is far from clear. What is the purpose for each step? More importantly, what kind of gas was used for testing? Different conclusions can be drawn when using different test gases.

Response:

Thanks for the comment. The experimental process was as following

The coals extracted from the well were processed to get the cube specimens, and then we started the experiment. The experimental steps are as follows:

(1) Place the specimen on the loading plate, and then put on a heat-shrinkable tube, which should cover the sealing washer of the loading plate (Fig. R1). The heat-shrinkable tube is tightly attached to the surface of the specimen with a hot air gun. Then the heat-shrinkable tube is tightly fastened at the sealing washer to prevent air leakage during the experiment.

(2) The specimen is fed into the loading cell, and the LVDT is installed and connected. The force of the true triaxial control system is cleared and the loading plate is moved slowly towards the specimen by displacement control. Meanwhile, the indicator changes

of the LVDT displayed on the system are observed. When the system shows that the force of the six loading plates is 0.5 kN to 2.5 kN (indicating that the loading plate has been in contact with the specimen) and the LVDT is within -1 mm ~ +1mm (the range of LVDT is -5 mm ~ +5mm), the installation of the LVDT is feasible. Otherwise, the LVDT will be readjusted (Measurements of the specimen's deformation are made independently along each principal stress axis using the linear variable displacement transducers as shown in Fig. R2).

(3) Seal the loading cell and fill it with hydraulic oil. When it is full, close the intake valve. The hydraulic oil was pressurized to 5 MPa by the servo valve (according to the principle of force and reaction force, the stress designed by the experiment is correspondingly reduced by 5 MPa) and the oil pressure should be greater than the gas pressure in the experiment to prevent the heat-shrinkable tube from bursting. Then enter the gas and adjust the pressure valve to 3 MPa.

(4) The experiment is carried out according to the designed stress path. When the stress reaches each stress point, the change of the LVDT and the flow rate of the flowmeter are observed. The next step can only be taken after the displacement and flow are stable.

We conducted the experimental scheme of loading and unloading the intermediate principal stress while fixing the major and minor principal stresses by considering the relative positions of principal stress and bedding direction. The differences among the permeability evolution and deformation properties in all directions were discussed to investigate the bedding effect. This was the purpose of each step.

The gas environment was CO₂ with 3 MPa in the experiment.

Comment 5: Line 159-160: it is impossible to get the conclusion from Fig. 5.

Response:

Thanks for the comment. According to the counting rule of the multi-functional true triaxial geophysical apparatus system, the strain increases when the specimen is compressed in one direction, and decreases when it expands. Figure 5 in original manuscript shows the variation of ε_v and k during the entire experimental process. ε_v and k basically comply to the following rules: with the increase of ε_v , a decrease in k induced by the compaction and closure of micro-porosities and microcracks occurred, and vice versa. Similarly, Yin et al. also obtained similar conclusions in the conventional triaxial test [12].

Reference:

[12] Yin G, Jiang C, Wang J G, Xu J. Geomechanical and flow properties of coal from loading axial stress and unloading confining pressure tests. *International Journal of Rock Mechanics & Mining Sciences*, 2015; 76:155-161. (doi:10.1016/j.ijrmms.2015.03.019)

Comment 6: In Conclusion: "the strain variation was largest, regardless of loading or unloading in the direction vertical to bedding". This conclusion is incorrect when it comes to coal if using non-sorptive gas as the test gas.

Response:

Thanks for the comment. We have revised this conclusion to 'The conclusion 'the strain variation was largest of coal saturating CO₂ under applied stress and keeping gas

pressure unchanged , regardless of loading or unloading in the direction vertical to bedding’.

Comment 7: No new findings are reported in this study.

Response:

Thanks for the comment. Coal is a kind of sedimentary rock with obvious beddings [13], especially for normal coals [14]. The existence of beddings weaken the continuity and integrity of coal and makes it heterogeneity [15, 16, 17]. With respect to underground reservoirs, the in situ stress is generally anisotropic ($\sigma_1 > \sigma_2 > \sigma_3$), and the hydrostatic stress and triaxial stress (i.e. varying confining stress) do not replicate the true in situ stress [18, 19, 20, 21]. Therefore, we performed experiments of cubic coal samples to investigate the permeability evolution and mechanical behavior of coal under true triaxial stress conditions by using newly developed true triaxial geophysical (TTG) apparatus.

The results show that volumetric strain, stress states and bedding directions determine the permeability comprehensively. The variable quantity of strain was the largest in the direction normal to the bedding plane. The expansion or compression degree were characterized by the difference between the major and minor principal strain ($\varepsilon_1 - \varepsilon_3$). Essentially, this represents the difficulty degree with regard to coal being compressed at the initial stress state and the deformation degree in ε_1 and ε_3 direction. Under an identical true triaxial stress condition, permeability was smaller when larger stress was applied in the direction normal to the bedding plane. Additionally, stress level in the direction parallel to the bedding planes and the directions between stresses in the direction

paralleling to the bedding planes and the flow direction also affect the permeability and strain.

This study can more realistically reflect the permeability and deformation evolutions of coals containing bedding under true triaxial conditions.

Reference:

- [13] Wang Z, Pan J, Hou Q, Yu B, Li M, Niu Q. Anisotropic characteristics of low-rank coal fractures in the Fukang mining area, China. *Fuel*, 2018, 211:182-193. (doi:10.1016/j.fuel.2017.09.067)
- [14] Cao Y, Mitchell G D, Davis A, Wang D. Deformation metamorphism of bituminous and anthracite coals from China. *International Journal of Coal Geology*, 2000, 43(1):227-242. (doi:10.1016/s0166-5162(99)00061-0)
- [15] Seedsman R W. Geotechnical sedimentology—its use in underground coal mining. *International Journal of Coal Geology*, 2001, 45(2):147-153. (doi:10.1016/s0166-5162(00)00029-x)
- [16] De Silva G P D, Ranjith P G, Perera M S A, Chen B. Effect of bedding planes, their orientation and clay depositions on effective re-injection of produced brine into clay rich deep sandstone formations: Implications for deep earth energy extraction. *Applied Energy*, 2016, 161:24-40. (doi:10.1016/j.apenergy.2015.09.079)
- [17] Kutchko B G, Goodman A L, Rosenbaum E, Natesakhawat S, Wagner K. Characterization of coal before and after supercritical CO₂ exposure via feature relocation using field-emission scanning electron microscopy. *Fuel*, 2013, 107, 777–786. (doi:10.1016/j.fuel.2013.02.008)

- [18] Li M, Yin G, Xu J, Li W, Song Z, Jiang C. A novel true triaxial apparatus to study the geomechanical and fluid flow aspects of energy exploitations in geological formations. *Rock Mechanics & Rock Engineering*, 2016; 49(12):4647-4659. (doi:10.1007/s00603-016-1060-7)
- [19] Li M, Yin G, Xu J, Cao J, Song Z. Permeability evolution of shale under anisotropic true triaxial stress conditions. *International Journal of Coal Geology*, 2016; 165:142-148. (doi:10.1016/j.coal.2016.08.017)
- [20] Burra A, Esterle J S, Golding S D. Horizontal stress anisotropy and effective stress as regulator of coal seam gas zonation in the Sydney Basin, Australia. *International Journal of Coal Geology*, 2014, 132(1):103-116. (doi:10.1016/j.coal.2014.08.008)
- [21] Liu Y, Li M, Yin G, Zhang D, Deng B. Permeability evolution of anthracite coal considering true triaxial stress conditions and structural anisotropy. *Journal of Natural Gas Science & Engineering*, 2018, 52:492-506. (doi:10.1016/j.jngse.2018.02.014)

Appendix B

Responses to reviewers and editors

Manuscript Number: RSOS-181438.R1

Paper Title: Influence of Principal Stress Effect on Deformation and Permeability of Coal Containing beddings under True Triaxial Stress Conditions

Authors: Jiahui Dai, Chao Liu, Minghui Li and Zhenlong Song

Dear editors and reviewers,

Special thanks to the anonymous reviewers for their careful reviews and valuable comments. We have carefully checked these comments and made necessary corrections and modifications. We tried our best to improve the manuscript.

Improvements and clarifications have been made in the current manuscript. These changes are marked by **red words**.

We appreciate for the editors and reviewers' warm work earnestly, and hope that the manuscript after the corrections can satisfy the requirements for publication. Once again, thank you very much for your comments and suggestions.

Point-by-point Responses to the reviewers' comments

Reviewer #1:

Comment 1: Formatting error: page 190-194.

Response:

Thanks for the comment. We are very sorry for it. We have revised the formatting in revised manuscript.

Comment 2: In all the figures, some more information can be added to the caption mentioning what does in illustrates.

Response:

Thanks for the comment. We have added more information of caption in figures.